# Active sensing in the categorization of visual patterns

**Scott Cheng-Hsin Yang[1]\*, Máté Lengyel[1,2]†, Daniel M Wolpert[1]†**

[1]Computational and Biological Learning Lab, Department of Engineering, University of Cambridge, Cambridge, United Kingdom; [2]Department of Cognitive Science, Central European University, Budapest, Hungary

**Abstract** Interpreting visual scenes typically requires us to accumulate information from multiple locations in a scene. Using a novel gaze-contingent paradigm in a visual categorization task, we show that participants' scan paths follow an active sensing strategy that incorporates information already acquired about the scene and knowledge of the statistical structure of patterns. Intriguingly, categorization performance was markedly improved when locations were revealed to participants by an optimal Bayesian active sensor algorithm. By using a combination of a Bayesian ideal observer and the active sensor algorithm, we estimate that a major portion of this apparent suboptimality of fixation locations arises from prior biases, perceptual noise and inaccuracies in eye movements, and the central process of selecting fixation locations is around 70% efficient in our task. Our results suggest that participants select eye movements with the goal of maximizing information about abstract categories that require the integration of information from multiple locations.

**\*For correspondence:** schy2@eng.cam.ac.uk

†These authors contributed equally to this work

**Competing interests:** The authors declare that no competing interests exist.

## Introduction

Several lines of evidence suggest that humans and other animals direct their sensors (e.g. their eyes, whiskers, or hands) so as to extract task-relevant information efficiently (*Yarbus, 1967*; *Kleinfeld et al., 2006*; *Lederman and Klatzky, 1987*). Indeed, in vision, the pattern of eye movements used to scan a scene depends on the type of information sought (*Hayhoe and Ballard, 2005*; *Rothkopf et al., 2007*), and has been implied to follow from an active strategy (*Najemnik and Geisler, 2005*; *Renninger et al., 2007*; *Navalpakkam et al., 2010*; *Nelson and Cottrell, 2007*; *Toscani et al., 2013*; *Chukoskie et al., 2013*) in which each saccade depends on the information gathered about the current visual scene and prior knowledge about scene statistics. However, until now, studies of such active sensing have either been limited to search tasks or to qualitative descriptions of the active sensing process. In particular, no studies have shown whether the information acquired by each individual fixation is being optimized. Rather, the fixation patterns have either been described without a tight link to optimality (*Ballard et al., 1995*; *Epelboim and Suppes, 2001*) or compared to an optimal strategy only through summary statistics such as the total number of eye movements and the distribution of saccade vectors (*Najemnik and Geisler, 2005*; *2008*) that could have arisen through a heuristic. Therefore, these studies leave open the question as to what extent eye movements truly follow an active optimal strategy. In order to study eye movements in a more principled quantitative manner, we estimated the efficiency of eye movements in a high-level task on a fixation-by-fixation basis.

Here, we focus on a pattern categorization task that is fundamentally different from visual search, in which often there is a single location in the scene that has all the necessary information (the target), and eye movements are well described by the simple mechanism of inhibition of return (*Klein, 2000*). In contrast, in many other tasks, such as constructing the meaning of a sentence of

**eLife digest** To interact with the world around us, we need to decide how best to direct our eyes and other senses to extract relevant information. When viewing a scene, people fixate on a sequence of locations by making fast eye movements to shift their gaze between locations. Previous studies have shown that these fixations are not random, but are actively chosen so that they depend on both the scene and the task. For example, in order to determine the gender or emotion from a face, we fixate around the eyes or the nose, respectively.

Previous studies have only analyzed whether humans choose the optimal fixation locations in very simple situations, such as searching for a square among a set of circles. Therefore, it is not known how efficient we are at optimizing our rapid eye movements to extract high-level information from visual scenes, such as determining whether an image of fur belongs to a cheetah or a zebra.

Yang, Lengyel and Wolpert developed a mathematical model that determines the amount of information that can be extracted from an image by any set of fixation locations. The model could also work out the next best fixation location that would maximize the amount of information that could be collected. This model shows that humans are about 70% efficient in planning each eye movement. Furthermore, it suggests that the inefficiencies are largely caused by imperfect vision and inaccurate eye movements.

Yang, Lengyel and Wolpert's findings indicate that we combine information from multiple locations to direct our eye movements so that we can maximize the information we collect from our surroundings. The next challenge is to extend this mathematical model and experimental approach to even more complex visual tasks, such as judging an individual's intentions, or working out the relationships between people in real-life settings.

written text, or judging from a picture how long a visitor has been away from a family (*Yarbus, 1967*), no single visual location has the necessary information in it and thus such tasks require more complex eye movement patterns. While some basic-level categorization tasks can be solved in a single fixation (*Thorpe et al., 1996*; *Li et al., 2002*), many situations require multiple fixations to extract several details at different locations to make a decision. Therefore, when people have to extract abstract information (eg., how long the visitor has been away) they need to integrate a series of detailed observations (such as facial expression, postures and gestures of the people) across the scene, relying heavily on foveal vision information with peripheral vision playing a more minor role (*Levi, 2008*).

We illustrate the key features of active sensing in visual categorization by a situation that requires the categorization of an animal based on its fur that is partially obscured by foliage (*Figure 1*). As each individual patch of fur can be consistent with different animals, such as a zebra or a cheetah, and the foliage prevents the usage of gist information (*Oliva and Torralba, 2006*), multiple locations have to be fixated individually, and the information accumulated across these locations, until a decision can be made with high confidence. For maximal efficiency, this requires a closed loop interaction between perception, which integrates information from the locations already fixated with prior knowledge about the prevalence of different animals and their fur patterns, and thus maintains beliefs about which animal might be present in the image, and the planning of eye movement, which should direct the next fixation at a location which has potentially the most information relevant for the categorization (*Figure 1*). Inspired by this example, and to allow a mathematically tractable quantification of the information at any fixation location, we designed an experiment with visual patterns that were statistically well-controlled and relatively simple, while ensuring that foveal vision would dominate by using a gaze-contingent categorization task in which we tracked the eye and successively revealed small apertures of the image at each fixation location. In contrast to previous studies (*Najemnik and Geisler, 2005*; *Renninger et al., 2007*; *Peterson and Eckstein, 2012*; *Morvan and Maloney, 2012*), our task required multiple locations to be visited to extract information about abstract pattern categories.

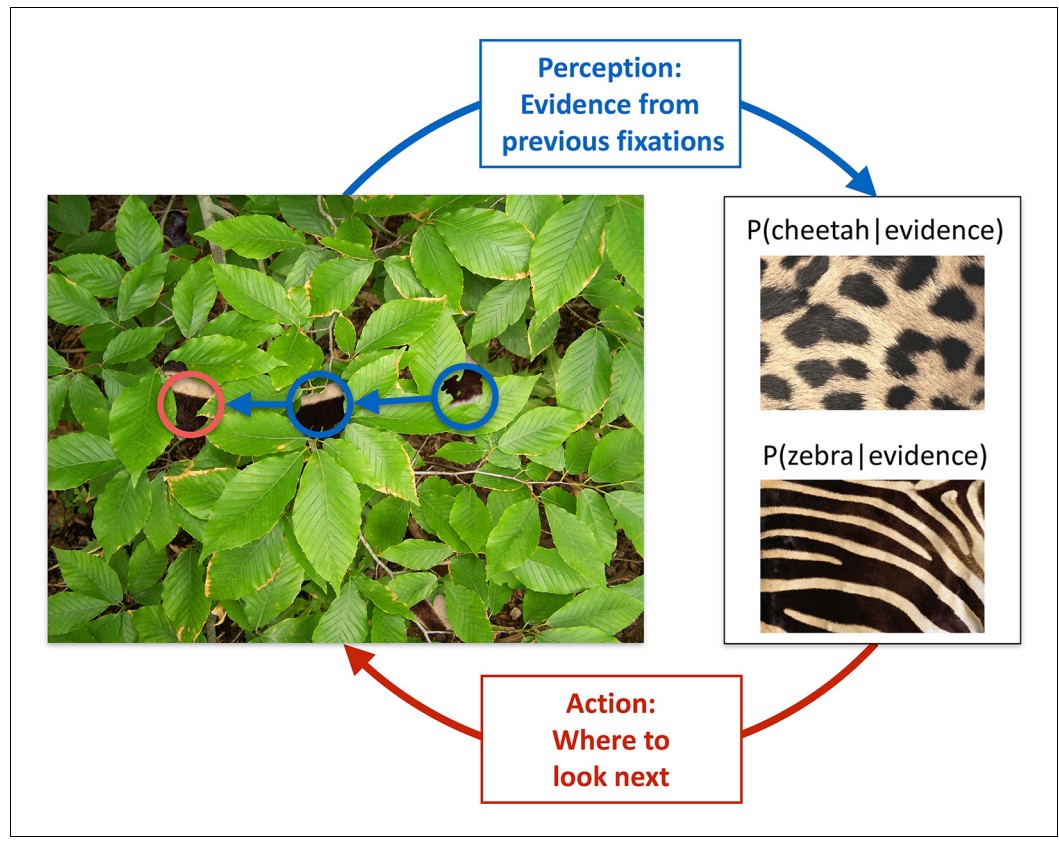

**Figure 1.** Active sensing involves an interplay between perception and action. When trying to categorize whether a fur hidden behind foliage (left) belongs to a zebra or a cheetah, evidence from multiple fixations (blue, the visible patches of the fur, and their location in the image) needs to be integrated to generate beliefs about fur category (right, here represented probabilistically, as the posterior probability of the particular animal given the evidence). Given current beliefs, different potential locations in the scene will be expected to have different amounts of informativeness with regard to further distinguishing between the categories, and optimal sensing involves choosing the maximally informative location (red). In the example shown, after the first two fixations (blue) it is ambiguous whether the fur belongs to a zebra or a cheetah, but active sensing chooses a collinearly located revealing position (red) which should be informative and indeed reveals a zebra with high certainty. Note that this is just an illustrative example.

## Results

### Categorization performance and eye movement patterns

We generated images of three types: patchy, horizontal stripy, and vertical stripy (*Figure 2A*). Participants had to categorize each image pattern as patchy or stripy (disregarding whether a stripy image was horizontal or vertical—the inclusion of two different stripy image types prevented participants from solving the task based on one image axis alone). The images were generated by a flexible statistical model that could generate many examples from each of the three image types, so that the individual pixel values varied widely even within a type and only higher order statistical information (ie. the length scale of spatial correlations) could be used for categorization. We first presented the participants with examples of full images to familiarize them with the statistics of the image types and to ensure their categorization with full images was perfect. We then switched to an active gaze-contingent mode in which the entire pattern was initially occluded by a black mask and the underlying image was revealed with a small aperture at each fixation location (*Figure 2B*; for visibility, the black mask is shown as white). As a control, we also used a number of passive revealing conditions in which the revealing locations were chosen by the computer rather than in a gaze-contingent manner. In all conditions, we controlled the number of revealings on each trial before requiring the

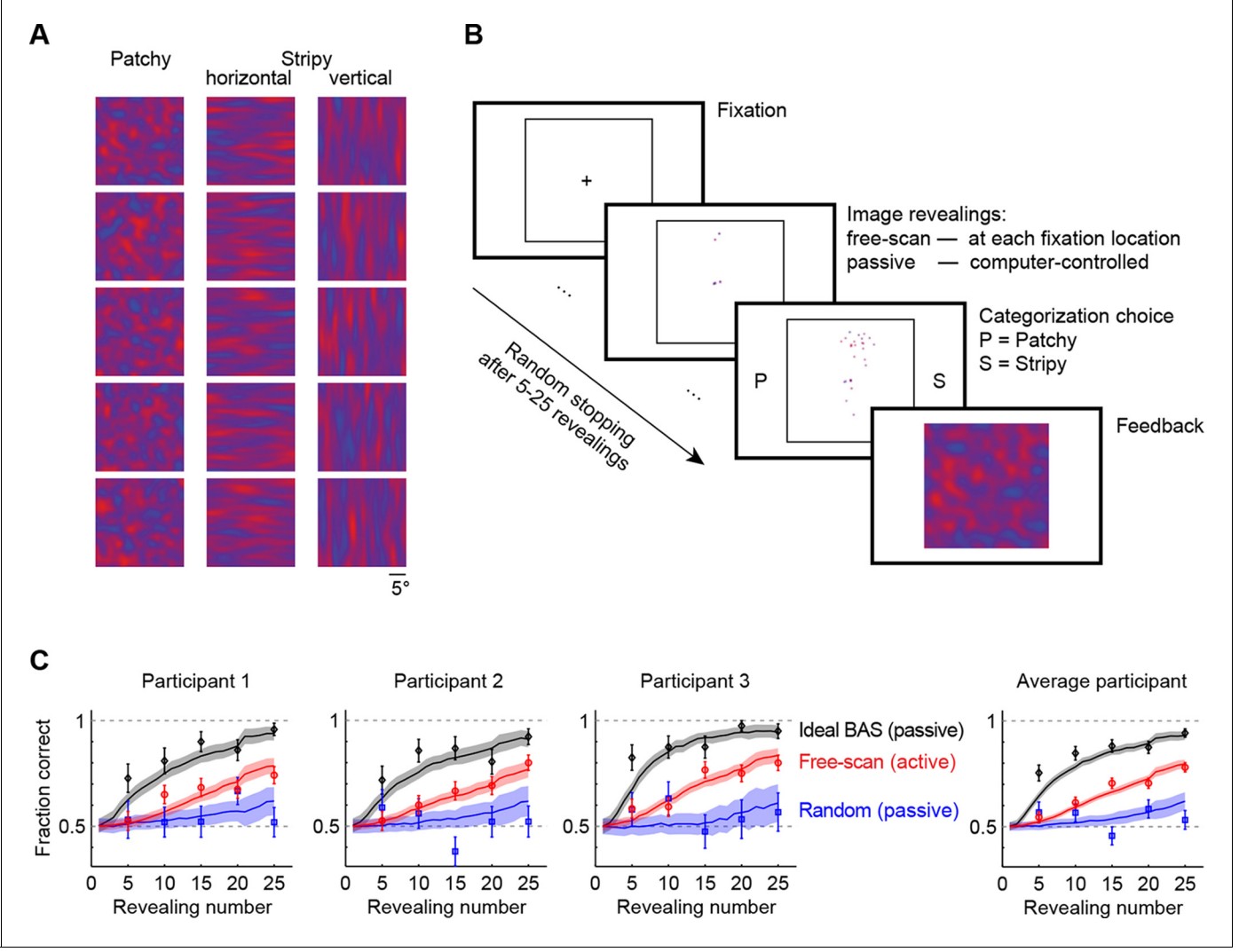

**Figure 2.** Image categorization task and participants' performance. (A) Example stimuli for each of the three image types sampled from two-dimensional Gaussian processes. (B) Experimental design. Participants started each trial by fixating the center cross. In the free-scan condition, an aperture of the underlying image was revealed at each fixation location. In the passive condition, revealing locations were chosen by the computer. In both conditions, after a random number of revealings, participants were required to make a category choice (patchy, P, versus stripy, S) and were given feedback. (C) Categorization performance as a function of revealing number for each of the three participants (symbols and error bars: mean ± SEM across trials), and their average, under the free-scan and passive conditions corresponding to different revealing strategies. Lines and shaded areas show across-trial mean ± SEM for the ideal observer model. *Figure 2—figure supplement 1* shows categorization performance in a control experiment in which no rescanning was allowed.

The following figure supplement is available for figure 2:

**Figure supplement 1.** Performance in the no-rescanning control experiment.

participants to categorize the image (*Figure 2B*). To ensure that participants had equal chance to extract information from all revealing locations in the passive as well as the active conditions we allowed the participants to rescan the revealed locations after the final revealing (see also Materials and methods for full rationale). Importantly, in the active revealing condition, even though rescanning was allowed after the final revealing, participants had to select all revealing locations in real time without knowing how many revealings they would be allowed on a given trial. Therefore, although rescanning could improve categorization it was unlikely to influence participants' active revealing strategy. To confirm this we also performed a control in which rescanning was not allowed (see below).

In the free-scan (active) condition, categorization performance improved with the number of revealings for each participant (*Figure 2C*, red points), indicating the successful integration of information across a large number of fixations. Although we used a gaze-contingent display, the task still allowed participants to employ natural every-day eye movement strategies, consistent with the inter-saccadic intervals (mean 408 ms) and relative saccade size (0.75–3.9 normalized by the three different relevant length scales of the stimuli) that were similar though somewhat shorter than those recorded for everyday activities (e.g. tea making; mean inter-saccadic interval 497 ms, and 0.95–19 relative saccade size normalized by the size of different fixated objects; *Hayhoe et al., 2003*; *Land and Tatler, 2009*).

To examine whether fixation locations depended on the underlying image patterns, we constructed revealing density maps for each image type. To account for the translation-invariant statistics of the underlying images of a type, we used the relative location of revealings obtained by subtracting from the absolute location of each revealing (measured in screen-centered coordinates) the center of mass of the absolute locations within each trial (Materials and methods). In order to compare eye movement patterns across conditions and participants, we subtracted the mean density map across all images for each participant. Importantly, we found that the pattern of revealings strongly depended on image type (*Figure 3A*, first four rows). The pattern of eye movements for images of the same type were positively correlated for each participant (*Figure 3B*, left, orange bars; $p<0.001$ in all cases), whereas eye movements for images of different types were negatively correlated (*Figure 3B*, left, purple bars; $p<0.001$ in all cases). We also found that eye movement patterns became increasingly differentiated over the course of the trial as progressively more of the image was revealed (*Figure 3B*, left, curves). The dependence of the eye movement patterns on the underlying image type shows that participants employed an active sensing strategy.

To assess whether this active strategy used by our participants contributed to performance improvement, we examined the same participants in a passive revealing condition. When the revealings were drawn randomly from an isotropic Gaussian centered on the image, performance was substantially impaired (*Figure 2C*, blue points), indicating that participants' decision performance benefited from their active sensing strategy. After the final revealing, participants rescanned only briefly, on average for 5.0 s in the active condition, 6.4 s in the passive random condition before making a decision. Moreover, their performance did not improve with increased rescanning time, instead it correlated negatively with rescanning time for 2 out of 3 participants, such that the probability of a correct decision for rescanning times at the 25th and 75th percentiles of the rescanning time distribution fell from 0.77 to 0.54 ($p<0.001$) and from 0.77 to 0.66 ($p<0.03$), respectively. This suggests that longer rescanning times indicate when participants are uncertain rather than providing the main source of information for their decisions. This is in contrast to additional revealings during the original scanning period which clearly benefit performance when chosen appropriately (*Figure 2C*).

To further examine whether the rescanning period after the final revealing affected the participants scanning strategy, we examined additional participants in a free-scan condition in which no rescanning was allowed after the final revealing (the display blanked). The revealing density maps for these participants were very similar to the maps of participants who were allowed to rescan (average within-type vs. across-type correlation across the two groups of participants: 0.63 vs. −0.30) and performance was also similar (*Figure 2—figure supplement 1* and *Figure 3—figure supplement 1*), although as expected slightly worse without rescanning. The proportion correct across all trials for the participants who were allowed to rescan was 0.65, 0.66 and 0.69 (average 0.66) and for those not allowed to rescan was 0.64, 0.58 and 0.66 (average 0.63). This confirms that allowing rescanning did not substantially change participants' revealing strategy.

## Bayesian ideal observer

We constructed an ideal observer (*Geisler, 2011*) which computed a posterior distribution, $\mathbb{P}(c|D)$, over image category $c$ given the observations $D$ (collection of previous revealing locations and revealed pixel values in the trial) and made a choice such that the category with higher posterior probability was more likely to be selected (see Materials and methods). To construct this model, we considered three sources of suboptimality: *prior biases* implying imperfect knowledge of the precise correlation length scales of each pattern category, *perception noise* that distorts the displayed pixel values, and *decision noise* which occasionally results in selecting the category with the lower

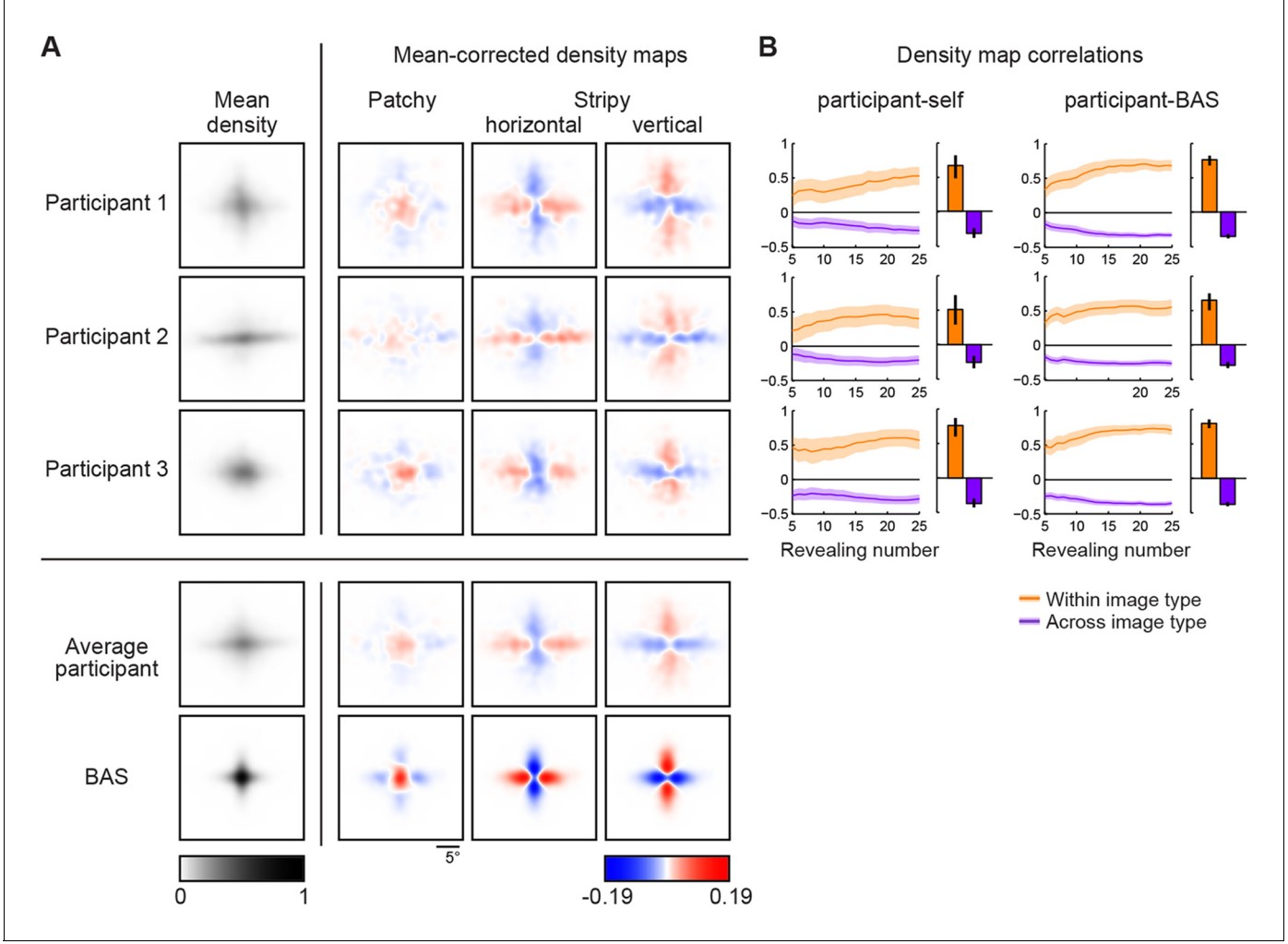

**Figure 3.** Density maps of relative revealing locations and their correlations. (**A**) Revealing density maps for participants and BAS. Last three columns show mean-corrected revealing densities for each of the three underlying image types (removing the mean density across image types, first column). Bottom: color scales used for all mean densities (left), and for all mean-corrected densities (right). All density maps use the same scale, such that a density of 1 corresponds to the peak mean density across all maps. *Figure 3—figure supplement 1* shows revealing density maps obtained for participants in a control experiment in which no rescanning was allowed. *Figure 3—figure supplement 2* shows the measured saccadic noise that was incorporated into the BAS simulations. *Figure 3—figure supplement 3* shows density maps separately for correct and incorrect trials. (**B**) The curves are correlations for individual participants as a function of revealing number with their own maps (left) and the maps generated by BAS (right). The bars are correlations at 25 revealing (see Materials and methods). Orange shows within image type correlation, ie. correlation between revealing densities obtained for images of the same type, and purple shows across image type correlation. Data are represented as mean ± SD for the curves and mean ± 95% confidence intervals for the bars.

The following figure supplements are available for figure 3:

**Figure supplement 1.** Revealing density maps in the no-rescanning control experiment.

**Figure supplement 2.** Saccadic variability and bias.

**Figure supplement 3.** Revealing density maps of the average participant in the main experiment split into correct and incorrect trials, and the BAS revealing maps.

posterior probability (*Houlsby et al., 2013*). We fitted six models to the individual choices of each participant. These models differed in the kind of prior bias and decision noise they included, and we selected the model with the strongest statistical evidence, as quantified by the Bayesian information criterion (*Tables 1–2*). The best model (4 parameters in total) provided a close match to the

**Table 1.** Maximum likelihood parameters of the model (see Materials and methods for details) with the best BIC score (see Table 2).

| Participant | Perception noise, $\sigma_P$ | Prior bias, $\Delta$ | Decision noise | |
|---|---|---|---|---|
| | | | Stimulus-dependent, $\beta$ | Stimulus-independent, $\kappa$ |
| 1 | 0.5 | 0.58° | 1.4 | 0.044 |
| 2 | 0.5 | 0.61° | 1.9 | 0.12 |
| 3 | 0.3 | 0.54° | 1.5 | 0.10 |

participants' performance both in the active and passive random-revealing conditions (*Figure 2C*, red and blue lines). Crucially, this model also allowed us to estimate the beliefs that participants held about image categories at any point in a trial, which was necessary for determining the optimal next eye movement that could maximally disambiguate between the categories.

## Predicting eye movement patterns by a Bayesian active sensor algorithm

To be able to rigorously assess how close our participants were to optimal sensing, we developed a Bayesian active sensor (BAS) algorithm which is optimal in minimizing categorization error with every single revealing. That is, for our task, the aim of BAS is to choose the next fixation location, $x^*$, so as to maximally reduce uncertainty in the category (*MacKay, 1992*). This objective is formalized by the BAS score function which expresses the expected information gain when choosing $x^*$, and which can be conveniently computed as:

$$\text{Score}(x^*|D) = \text{H}[z^*|x^*, D] - \langle \text{H}[z^*|x^*, c, D]\rangle_{\mathbb{P}(c|D)} \tag{1}$$

where $\text{H}$ denotes entropy (a measure of uncertainty), $z^*$ is the possible pixel value at $x^*$, $D$ is the collection of revealing locations and revealed pixel values that have been observed in the trial as above, $c$ is image category, and $\langle \cdot \rangle$ denotes averaging over the two categories weighted by their posterior probabilities, as computed by the ideal observer (for more details, see Materials and methods). This expresses a trade-off between two terms. The first term encourages the selection of locations, $x^*$, where we have the most overall uncertainty about the pixel value, $z^*$, while the second term prefers locations for which our expected pixel value for each category is highly certain.

*Figure 4A* shows a sequence of fixations on a representative trial. On each fixation, the BAS score is computed for all possible positions (grayscale map) based on all previous fixation locations (green dots) and the pixel values revealed there. While the BAS algorithm would choose the position with the highest BAS score as the next fixation location (blue crosses), the participant might choose a different, suboptimal, fixation location (yellow circles). Nevertheless, the informativeness of most of the participant's fixation locations were very high as expressed by their information-percentile values (the percentage of putative fixation locations with lower BAS scores than the one chosen by the participant).

**Table 2.** Model comparison results using Bayesian information criterion (BIC, lower is better). Each row is a different model using a different combination of included (+) and excluded (–) parameters (columns, see Materials and methods for details). Last column shows BIC score relative to the BIC of the best model (number 4).

| Model | Perception noise, $\sigma_P$ | Prior bias | | Decision noise | | BIC |
|---|---|---|---|---|---|---|
| | | Scale, $\alpha$ | Offset, $\Delta$ | Stimulus-dependent, $\beta$ | Stimulus-independent, $\kappa$ | |
| 1 | + | – | – | + | – | 160 |
| 2 | + | – | – | + | + | 139 |
| 3 | + | – | + | + | – | 58 |
| 4 | + | – | + | + | + | 0 |
| 5 | + | + | – | + | – | 105 |
| 6 | + | + | – | + | + | 102 |

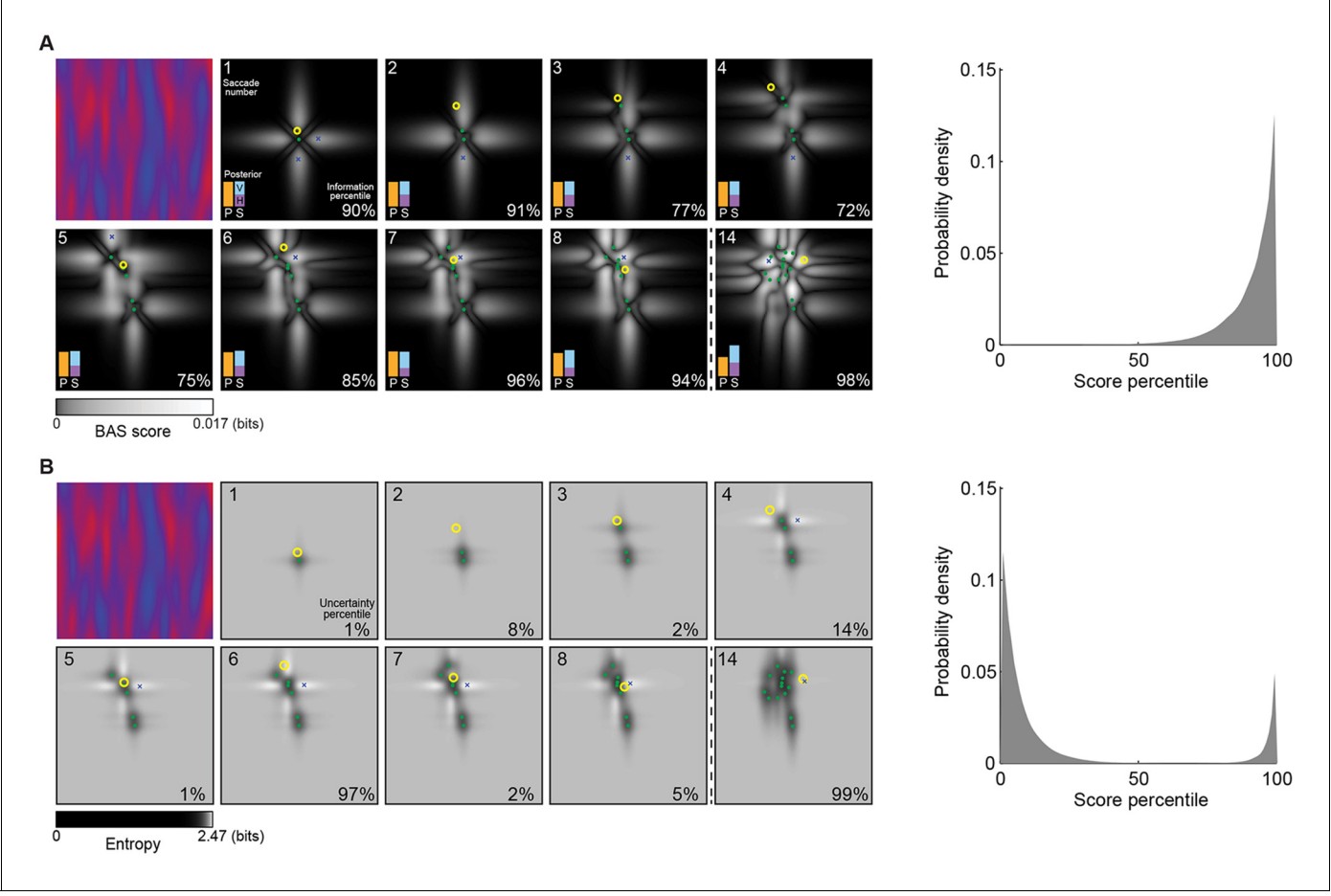

**Figure 4.** Example trial of the Bayesian active sensor (BAS) and its maximum entropy variant. (**A**) The operation of BAS in a representative trial for saccades 1–8 and 14 (underlying image shown top left). For each fixation (left, panels), BAS computes a score across the image (gray scale, *Equation 1*). This indicates the expected informativeness of each putative fixation location based on its current belief about the image type, expressed as a posterior distribution (inset, lower left), which in turn is updated at each fixation by incorporating the new observation of the pixel value at that fixated location. Crosses show the fixation locations with maximal score for each saccade, green dots show past fixation locations chosen by the participant and yellow circle shows current fixation location. Percentage values (bottom right) show their information percentile values (the percentage of putative fixation locations with lower BAS scores than the one chosen by the participant). Histogram on the right shows distribution of percentile values across all participants, trials and fixations. (**B**) Predictions of the maximum entropy variant (the first term in *Equation 1*) as in (**A**). For saccades 1–3, the fixation locations with maximal score (crosses) are not shown because the maxima comprise a continuous region near the edge of the image instead of discrete points. Note that entropy can be maximal further (eg. fixation 4) or nearer the edges of the image (eg. fixation 1), depending on the tradeoff between the two additive components defining it: the BAS score, which tends to be higher near revealing locations (panel A), and uncertainty due to the stochasticity of the stimulus and perception noise, which tends to be greater away from revealing locations. *Figure 4—figure supplement 1* shows two illustrative examples for this trade-off.

The following figure supplement is available for figure 4:

**Figure supplement 1.** Trade-off between the two components making up total uncertainty underlying the maximum-entropy algorithm.

We simulated eye movement patterns derived by BAS for the same images shown to our participants. In order to take into account basic biological constraints on the accuracy of eye movements, we included saccadic variability and bias in the model based on measurements made independently in a group of participants which took into account both the standard deviation and bias of saccades both along and orthogonal (standard deviation only) to the desired saccade direction as a function of desired amplitude (*Figure 3—figure supplement 2*). The predicted (mean-corrected) pattern of eye movements closely matched those observed (*Figure 3A*, last two rows): they were positively correlated with participants' eye movements for the same image type (*Figure 3B*, right, orange bars;

p<0.001 in all cases), but negatively correlated with those for different image types (*Figure 3B*, right, purple bars; p<0.001 in all cases). These differences increased as a function of revealing number (*Figure 3B*, right, curves). Moreover, when we split participants' trials into those in which they made a correct or incorrect decision, the pattern of eye movements derived from the correct trials correlated better with the BAS pattern than that derived from incorrect trials (*Figure 3—figure supplement 3*, $\rho_{\text{correct}} = 0.74$, $\rho_{\text{incorrect}} = 0.20$, p<0.001 for the average participant), further suggesting that following a BAS-like strategy was beneficial for performance.

For comparison, we also analyzed how well participants' fixations could be accounted for by a strategy using a variant of the score that only included the first term in Equation 1 and thus selected locations with maximal entropy rather than maximal information gain. We found that this strategy provided a substantially poorer fit to our eye movement data than the full BAS algorithm, as measured by the distribution of the scores corresponding to actual fixation locations (*Figure 4B*) and the anti-correlations between predicted and actual revealing maps at 25 revealings ($\rho = -0.62$, $-0.51$, $-0.43$; all p<0.001).

## Fixation informativeness

In order to obtain a fixation-by-fixation measure of the informativeness of participants' individual eye movements, we used the ideal observer model to quantify the amount of information accumulated about image category over subsequent revealings in a trial for different revealing strategies (*Figure 5A*). This information-based measure is more robust than directly measuring distances between optimal and actual scan paths because multiple locations are often (nearly) equally informative when planning the next saccade. For example, in the trial shown in *Figure 4A*, the BAS score map was clearly multimodal for several fixations, and some of the participant's fixation locations

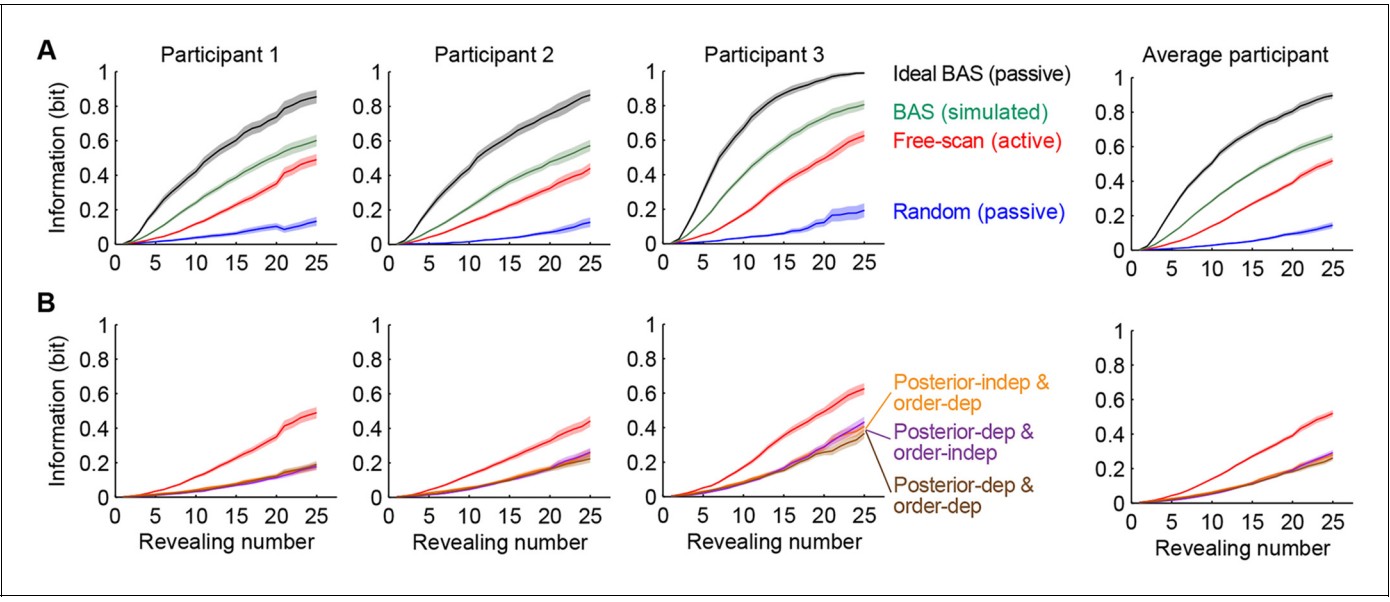

**Figure 5.** Information gain as a function of revealing number for different strategies. (**A**) Cumulative information gain of an ideal observer (matched to participants' prior bias and perceptual noise) with different revealing strategies (black, green, and blue) and participants' own revealings (red). Data are represented as mean ± SEM across trials. *Figure 5—figure supplement 1* shows a measure of efficiency extracted from these information curves across sessions. (**B**) Information gains for three heuristic strategies (See text for details, and Materials and methods): posterior-independent & order-dependent fixations (orange), posterior-dependent & order-independent fixations (purple), and posterior- & order-dependent fixations (brown). The information gain curves for the three heuristics overlap in all cases. Participants' active revealings (red lines, as in A) were 1.81 (95% CI, 1.68–1.94), 1.85 (95% CI, 1.72–1.99), and 1.92 (95% CI, 1.74–2.04) times more efficient in gathering information than these heuristics, respectively. Data are represented as mean ± SEM across trials.

The following figure supplement is available for figure 5:

**Figure supplement 1.** Relative efficiency across free-scan sessions.

were indeed distant from the corresponding locations that BAS would have chosen, yet their informativeness was generally very high. Therefore, categorization performance is better understood in terms of information measures on the revealed locations rather than by the geometry of individual scan paths, which has traditionally been used in previous studies (*Najemnik and Geisler, 2005*; *Renninger et al., 2007*).

We compared the information efficiency of different revealing strategies by measuring the relative number of revealings required by them to gain the same amount of information. Participants' active revealings (*Figure 5A–B*, red lines) were 2.93 (95% confidence interval [CI], 2.60–3.32) times more efficient than random revealings (*Figure 5A*, blue lines). As a more stringent control than a random strategy, we also simulated eye movement patterns for three heuristic strategies that reproduced different aspects of the statistics of participants' actual eye movement patterns but lacked the full closed-loop Bayesian interaction between belief updating and eye-movement selection (see Materials and methods for details). We first used a feed-forward strategy, which retained order-dependence, i.e. the way the statistics of revealing locations chosen by the participants depended on revealing number, but ignored participants' inferences about the category (*Figure 5B*, orange). The second heuristic took into account the participant's belief about the underlying image, but not the revealing number (*Figure 5B*, purple). The third heuristic was a partial closed-loop strategy, that thus respected both the belief- and order-dependence of revealings, but not the details of previous revealings (*Figure 5B*, brown line). Notably, this strategy would be optimal for simpler stimuli and tasks, such as visual search for a target, but not for our spatially correlated stimuli and task requiring information integration across multiple locations. Participants' revealings were 1.81–1.92 times more efficient than these three heuristic strategies.

However, participants' active revealings were less efficient by a factor of 2.48 (95% CI, 2.33–2.62) than the information provided by revealings generated by the BAS algorithm with 'idealized' parameters: minimal perception noise (2–3 times lower than our participants'; see Materials and methods) and no prior biases or saccadic inaccuracies (*Figure 5A*, black lines). Indeed, this efficiency in information gathering was also reflected in our participants' performance when they viewed revealings generated by this ideal BAS (*Figure 2C*, black points). In this condition, participants only rescanned the revealed locations for a short amount of time after the final revealing (average of 4.4 s), less than in the active and passive random conditions, thus their increased efficiency could not be attributed to longer rescanning times. The discrepancies between participants' and BAS's revealings may be caused by participants employing an inefficient strategy to select their fixation locations, or, alternatively, they may be due to more trivial factors upstream or downstream of the process responsible for selecting the next fixation, such as noise and variability in perception or execution, respectively. Importantly, when we computed the information gain provided by BAS when operating with participants' prior biases, perceptual noise and the typical saccadic inaccuracies as described above, the discrepancy between the informativeness of BAS-generated revealings compared to that of participants' revealings was markedly reduced (*Figure 5A*, green lines, BAS/free-scan efficiency = 1.45, 95% CI, 1.37–1.53). This suggests that the central component of choosing where to fixate was around 70% efficient in our participants, and a large component of suboptimality arose due to other processes. To examine the role of learning in participants' active sensing strategy, we computed the relative efficiency of each of the free-scan sessions of our experiment, spanning multiple days, compared to the final session (*Figure 5—figure supplement 1*). We found that efficiency remained stable over the whole course of the experiment, suggesting there was minimal, if any, learning required in the free-scan task.

## Discussion

Our results show that humans employ an active sensing strategy in deciding where to look in a high-level pattern categorization task. In our task, participants' patterns of eye movements were well predicted by a Bayes-optimal algorithm seeking to maximize information gain relevant to the task with each individual eye movement. This Bayes optimal strategy involved finding the location in the scene which when fixated was most likely to lead to the greatest reduction in categorization error, rather than simply the location associated with the most uncertainty about pixel value.

## The efficiency of active sensing in human vision

Although our participants performed better when revealings were chosen by the BAS algorithm than with their own scan paths, our results suggest that this suboptimality in participants' performance was to a large part due to prior biases, perception noise, and saccadic inaccuracies constraining the selection of fixation locations rather than an inefficient active sensing strategy. In particular, our participants' eye movements were substantially more efficient than heuristic strategies that only employed a subset of the elements of a fully closed-loop active sensing strategy, and were about 70% as efficient as the optimal active sensor that operated with the participants' own prior bias, perception noise, and saccadic inaccuracies. Importantly, this estimate does not conflate the inference process with the selection process in that even if the participant's inference is biased or inconsistent, we can measure, given those beliefs, the extent to which they select fixation locations optimally. The 30% inefficiency we found may be due to unmodeled constraints in human eye movement strategies, such as the biases for cardinal directions or fixating the centre of an image that were apparent in our data (*Figure 3A*) and that may also be beneficial in natural environments (*Tatler and Vincent, 2008*; *2009*). Such suboptimalities are conceptually different from those that we factored out using the specific biases and noise included in the BAS model. The latter are suboptimalities that arise in the execution of a planned saccade (as in *Figure 3—figure supplement 2*), while the former could be suboptimalities of the planning itself. Therefore, as we were interested in the degree of (sub)optimality of the planning component, we chose not to factor out potential biases for cardinal directions or locations. Should such suboptimalities turn out to be part of the execution process, then our estimate of 70% would become a lower bound on the efficiency of the planning process.

Our results contrast with recent studies suggesting that the pattern of eye movements do not follow an active sensing strategy. In one study, using a simple object localization task, it was found that the choice of fixation locations was close to random despite obvious learning of the underlying stimulus statistics (*Holm et al., 2012*). In another study, using a simplified visual search task, participants' eye movements were virtually unaffected by the configuration of the visual stimuli presented in the trial (*Morvan and Maloney, 2012*). In contrast, we found that participants used both their prior knowledge of stimulus statistics as well as evidence accumulated about the current visual scene to guide their eye movements. We speculate that better performance in our case may be due to our more naturalistic task, which involves the extraction of abstract latent features, revealing more typical processing of the sensorimotor system.

## Relevance for natural vision

Although our task was designed to emulate many features of natural vision, there remain several differences. First, our gaze-contingent display involved exposing small patches of the image at each fixation whereas in natural vision information is available from the full visual field, albeit with decreasing acuity away from the fovea. If participants' selection process was aware of and actively optimized for the changed field of vision in our task, the resulting eye movement patterns could be different from those in natural vision. Nevertheless, the basic summary statistics of (macro-)saccades in our task (average size and inter-saccadic intervals) were similar to those found in natural vision (*Hayhoe et al., 2003*; *Land and Tatler, 2009*), and there was a lack of adaptation to the task over the course of several days of free-scan sessions (*Figure 5—figure supplement 1*). These seem to suggest, at least indirectly, that participants did not depart dramatically from their natural eye movement strategies.

Second, our task focuses on the voluntary component of eye movements, that is the scan paths of fixations, rather than the more involuntary processes of micro-saccades and drift (*Rolfs, 2009*; *Ko et al., 2010*; *Rucci et al., 2007*; *Poletti et al., 2013*; *Kuang et al., 2012*), which did not trigger new revealings. Importantly, in our task these involuntary processes are likely to be used for extracting information within each revealed patch, whereas high-level categorization required macro-saccades. This is because these involuntary processes cover an area with a standard deviation (SD) on the order of 0.22° (over the course of each fixation; *Rolfs, 2009*), which is similar to our Gaussian apertures (SD of 0.18°), whereas the smallest length scale of our stimuli (0.91°) is 4 times larger. However, it is possible that in natural scenes with finer structure, micro-saccades and drift may contribute more to abstract feature extraction.

Third, our stimuli set had strictly stationary statistics. This is an approximation to natural image statistics that are often assumed to be spatially stationary on average (eg. *Field, 1987*) but usually include local non-stationarities (eg. different objects with different characteristic spatial frequencies). Nevertheless, our stimuli were more naturalistic than many of the stimuli used in active sensing studies of visual search (eg. simple 1/f noise) while still allowing a rigorous control and measurement of the amount of high-level category information available at any potential revealing location which would not have been possible with natural scenes.

Finally, natural vision works under time constraints. Our task limited the number of revealings in each trial rather than the time, although the temporal aspects of the eye movements (inter-saccadic intervals) were similar to eye movements with natural scenes. We also showed in a control experiment that the additional time allowed for rescanning the revealed locations did not affect the initial scanning strategy.

A challenge for future studies will be to employ visual tasks that are more naturalistic in these respects while still retaining our ability to quantify the task-relevant information available to participants at any point in a trial. It may well be that participants are more efficient in such naturalistic settings than the 70% we found in our task.

## Relation to earlier work

Our approach makes several important contributions. First, by using a gaze-contingent display we were able to isolate the top-down strategies of eye movement selection, thereby complementing studies which examined the influence of low-level visual features and salience (*Itti and Koch, 2000*; *Wismeijer and Gegenfurtner, 2012*) on eye movement selection. In addition, our task focuses on integrating low-level visual information into an abstract category thus complementing studies that examine the effect of target value (*Navalpakkam et al., 2010*; *Krajbich et al., 2010*; *Markowitz et al., 2011*; *Schutz et al., 2012*) or more cognitive tasks that use eye movement (*Nelson and Cottrell, 2007*) or button-pressing for information search (*Castro et al., 2009*; *Gureckis and Markant, 2009*; *Borji and Itti, 2013*). Some of these studies used a similar Bayesian formalism to obtain the active learning strategies specific to their tasks and showed that humans perform active learning when given the opportunity to consciously deliberate from which location in a scene they should gather information next. In contrast, our work shows that high-efficiency active sensing is a natural strategy for eye movements that humans adopt without overt deliberation, as evidenced by the inter-saccadic intervals matching naturalistic tasks and the fact that our participants reached near-asymptotic efficiency already in the first session of our task.

Second, by using a pattern categorization task we could ensure that no single location was especially informative but that the task required an integration of information from multiple locations both to select the next eye movement and to solve the task, and specifically that a closed-loop strategy was necessary for solving the task efficiently. This is in contrast with studies that attempted to quantify the general informativeness of single locations in a scene (*Gosselin and Schyns, 2001*), and showed that they are the target of fixations humans tend to choose in general (*Peterson and Eckstein, 2012*; *Toscani et al., 2013*; *Chukoskie et al., 2013*), or visual search in which, by the nature of the task, the target location is fully informative by itself (*Najemnik and Geisler, 2005*). As such, most previous studies have not addressed the important interplay between information gathering and fixation selection characteristic of naturalistic active sensing: that, in general, the most informative location is ever-changing, dependent on the history of fixations one has already performed.

Third, in contrast to active sensing for simple visual search (*Najemnik and Geisler, 2005*; *Navalpakkam et al., 2010*; *Morvan and Maloney, 2012*), our formalism extends the range of active sensing to tasks which have arbitrary, not necessarily spatial, latent features (such as categories). In particular, it provides the first fixation-by-fixation analysis of information gathering under active eye movements by carefully matching our observer model to participants' performance. As a result, for the first time, we were able to dissociate the contributions of the eye-movement selection process from those of perceptual, motor, and decision processes, identify predominant sources of apparent sub-optimality in active sensing, and quantify the efficiency of choosing each individual fixation throughout scanning a whole scene. Taken together, these features make our approach amenable to multiple tasks in different perceptual domains (*Kleinfeld et al., 2006*), as well as high-level cognitive tasks such as estimating the age or socio-economic status of people in a scene (*Yarbus, 1967*).

# Materials and methods

## Participants
Three naive participants (aged 25–35 years, none of them were authors or neuroscientists) took part in the experiment. All participants were neurologically healthy, had normal or corrected to normal vision and gave their informed consent before participating. The study was approved by the institutional ethics committee. Each experiment took approximately 12 hr across 6 days (2 hr per day). As this experiment was particularly laborious we focus on within-participant analysis.

## Experimental apparatus and setup
Participants sat 42 cm in front of a 17″ Sony Multiscan G200 FD Trinitron CRT monitor (32-bit color, 1024x768 resolution, 100 Hz refresh rate). An EyeLink 1000 eye tracker was used to track the participant's right eye position at 1000 Hz. A chin and forehead rest was used to stabilize the head.

## Stimuli
Stimuli were generated such that the value of a pixel, $z$, depended on its two-dimensional location, $x$, through a function $z = f(x)$ sampled from a two-dimensional Gaussian process (**Rasmussen and Williams, 2006**) with zero mean and covariance function $K_\theta(\cdot, \cdot)$. The covariance function was squared-exponential and it was parameterized by $\theta = \{\lambda_h, \lambda_v\}$ setting the image pattern's horizontal and vertical correlation length scales, with the variance set to unity, such that the covariance of the pixel values at two positions, $x$ and $x'$ (each two-dimensional), was:

$$K_\theta(x, x') = e^{-\frac{1}{2}(x-x')^T \begin{bmatrix} \lambda_h^2 & 0 \\ 0 & \lambda_v^2 \end{bmatrix}^{-1} (x-x')} \tag{2}$$

For the patchy ($\mathrm{PA}$), stripy horizontal ($\mathrm{SH}$), and stripy vertical ($\mathrm{SV}$) pattern types the hyperparameters were $\theta_{\mathrm{PA}} = \{1.39°, 1.39°\}$, $\theta_{\mathrm{SH}} = \{4.63°, 0.91°\}$, and $\theta_{\mathrm{SV}} = \{0.91°, 4.63°\}$, respectively. Function values ($z$) in the range $-4$ to $4$ were mapped to image pixel colors so that the extremes corresponded to pure red (RGB value [1, 0, 0]) and blue ([0, 0, 1]), with intermediate values being linearly interpolated in RGB space. Functions which had values outside $[-4, 4]$ were discarded and a new function was generated. The images were sampled over a grid of $77\times77$ locations subtending $27.8°$ at the eye in the horizontal and vertical dimensions and then supersampled, using 2D splines, up to a resolution of $770\times770$ pixels (this allowed images to be generated rapidly without compromising visual appearance). On each trial, a pattern category $c$, that was patchy ($\mathrm{P}$) or stripy ($\mathrm{S}$), was chosen with equal probability and if the stripy category was chosen then a horizontal or vertical pattern type was chosen with equal probability. Having two different types within the stripy pattern category ensured that the optimal scan path depended on the image (otherwise, always scanning in one direction where the length scales were different would be optimal).

## Task
The task on each trial was for participants to determine whether a pattern displayed on the monitor was patchy or stripy (irrespective of whether it was vertical or horizontal) under different experimental conditions. The experiment consisted of four conditions: training, free-scan familiarization, free-scan, and passive revealing.

### Training (8 sessions × 40 trials)
Participants triggered the start of each trial by fixating (within 1.5° for 500 ms) a cross centered on the screen, at which point the cross disappeared. An image was displayed centered on the location of the fixation cross, and participants had to decide whether the image was patchy or stripy. They were allowed to scan the image freely for up to 10 s and make their decision by fixating (within 2.8° for 800 ms) on one of the two choice letters, P or S, which were displayed to the left and right of the displayed image, respectively. Participants received audio feedback as to the correctness of their choice. Twenty images from each category were presented in a randomized order. The training sessions ensured that participants could learn the statistics of the image patterns. Categorization performance was perfect in training sessions for all participants.

### Free-scan familiarization (5 sessions × 40 trials)

Each trial started with participants fixating a center cross. A randomly generated image from one of the categories was then displayed but initially completely obscured by a black mask. Participants could freely scan the display, and wherever they fixated, the underlying image was revealed by unmasking a small aperture at the fixation location. This was achieved by revealing an isotropic Gaussian region with standard deviation 0.18° at the fixation location, with the values of the Gaussian used to interpolate linearly between complete transparency (alpha=0) at the maximum of the Gaussian and black (alpha=1) where the value of the Gaussian is 0. A new fixation was detected, and hence a new revealing triggered, when the following three criteria were met: 1) a saccade had occurred since the last fixation as determined by eye speed greater than $59°\ s^{-1}$; 2) the displacement from the previous fixation was greater than 0.59°; and 3) the standard deviation of the eye position was less than 0.28° for the last 100 ms. These parameters were based on pilot experiments with the aim of making the revealings in the free-scan session feel natural so that each location viewed was revealed and spurious locations were not revealed. Participants were required to make 25 fixations before making their decision. After the 25 revealings, the category choices appeared (P vs. S) and participants had 60 s to choose a category with the revealed locations remaining on the display. Upon answering, participants were shown the full image with audio feedback. All participants achieved an average performance of 70% accuracy or higher.

### Free-scan (6 sessions × 100 trials)

Free-scan trials were exactly the same as free-scan familiarization trials except that the number of revealings across trials was chosen randomly on each trial from 5 to 25 in steps of 5 (balanced) and was a priori unknown to the participants. The choice letters, P and S, appeared after the given number of revealings and no new revealings occurred after this point. The unknown, random stopping number of revealings served to encourage participants to be greedy in their information seeking.

### Passive revealing (8 sessions × 100 trials)

This session was the same as the free-scan session except that the revealing locations were predetermined by an algorithm independent of participant's eye movements and sequentially appeared at intervals of 400 ms (which was about the average interval between consecutive fixations in the free scan experiment: 408 ms, which in turn was very similar to the inter-saccadic intervals measured under natural viewing conditions in everyday tasks; *Hayhoe et al., 2003*; *Land and Tatler, 2009*). Participants were instructed to follow the revealings as much as possible and were allowed to scan the scene after all revealings had appeared until they made their category decision. The algorithm followed one of three strategies randomly chosen:

1. Random: revealing locations were drawn from a scene-centered isotropic Gaussian with standard deviation 9.27°. For comparison, participants' revealing locations in the free-scan condition had an average location that was 0.05° from the center of the scene and had a standard deviation 4.50°. Revealing locations that fell outside the image were resampled;
2. Ideal BAS: revealing locations were generated by the BAS algorithm (see below). For establishing an upper bound on the informativeness of the optimal revealing locations, the algorithm was allowed to access the displayed, as opposed to perception noise-corrupted pixel values, and did not include prior biases and saccadic inaccuracies (see below).
3. Anti-BAS: revealing locations were generated by the BAS algorithm as above but as if it was observing a different image than the real one, which belonged to a wrong type. For example, if the real image belonged to type SH, the revealing locations were generated based on an image from type PA or SV (randomly chosen).

We mixed the BAS and anti-BAS trials to ensure participants could not use the pattern of revealing locations (independent of pixel values) to infer the category of the underlying image. The experiment included 8 passive revealing sessions with a total of 200, 200, and 400 trials from strategy 1, 2, and 3, respectively. The trials were first randomly mixed then divided into 8 sessions.

The eye tracker was calibrated before each session (25-point calibration for the free-scan condition and 9-point calibration for the passive-revealing conditions). Drift correction was performed at the start of each trial after fixation on the center cross was achieved. Re-calibration was performed

whenever participants reported that the revealing locations did not match where they fixated, whenever they could not trigger the start of a trial, or make their category choice by fixating.

Participants ran using the following schedule: day one, 3 training sessions intermixed with 5 free-scan familiarization sessions; day two and three, 1 training session and 3 free-scan sessions each; day four to six, 1 training session and 2 or 3 passive revealing sessions each. All the free-scan sessions came before any passive revealing sessions so as to avoid influencing participants' choice of eye movements by our choice of passive revealing strategies.

As we wished to compare the active strategy with passive revealing we allowed participants to rescan the revealed locations after the final revealing but before making a decision. This was critical as in the passive revealing conditions, although participants may detect and follow the revealings, because they are small and dispersed, they need to scan the scene to find and view them all. Therefore, the reason for allowing additional time after the final revealing was so as to make sure they had a chance to extract as much information as they liked from the revealed locations (locations and pixel values). In order to make the conditions directly comparable, we followed the same procedure in the active condition. Thus, allowing a rescanning period was the only way we could make a fair comparison across conditions using our gaze contingent design, which in turn allowed fine control over the information participants could obtain by eye movements.

Crucially, our key interest was in where participants chose to fixate during the initial revealing period and not in the perception model. During the active condition, all the selection happened prior to the rescanning phase and participants did not know how many saccades they would be allowed so they still needed to be efficient in choosing revealing locations even if they could rescan them later. Therefore, as we were analyzing the initial scanning, the final rescanning simply equalized the information that could be extracted from each revealing for all conditions but it was unlikely to influence the initial selection. Although, it is theoretically possible that participants adopted a different eye movement strategy knowing they could freely re-visit already revealed locations, the rescanning control (described below) suggests that this was not the case (*Figure 3—figure supplement 1*).

To examine whether rescanning time had an effect on performance, we fit each participants choice accuracy as a logistic function (bounded between 0.5 and 1) of rescanning time. We allowed different shifts per condition (active, passive random, passive BAS) but the same slope parameter across conditions.

## No-rescanning control

To directly examine whether the rescanning period allowed after the final revealing affected participants scanning strategy, we performed a control experiment with three additional naive participants. These participants performed the training, free-scan familiarization, and free-scan sessions as in the original experiment except that no rescanning after the final revealing was allowed. That is all revealings disappeared (i.e. display returned to a black screen) 350 ms after the saccade away from the final revealing and participants were required to indicate their choice.

## The ideal observer model of the task

The ideal observer maintains and continually (after each observation) updates a posterior distribution over categories, $c \in \{\mathrm{P}, \mathrm{S}\}$, given knowledge of the parameters defining each image type, $\theta \in \{\theta_{\mathrm{PA}}, \theta_{\mathrm{SV}}, \theta_{\mathrm{SH}}\}$, and data, $D = \{\mathbf{z}, \mathbf{x}\}$, which is the set of perceived pixel values $\mathbf{z} = \{z_1, \ z_2 \dots z_L\}$ at the $L$ locations $\mathbf{x} = \{x_1, \ x_2 \dots x_L\}$ revealed in the trial so far:

$$\mathbb{P}(c = \mathrm{P}|D) = \frac{\mathbb{P}(D|\theta_{PA})}{\mathbb{P}(D|\theta_{PA}) + \frac{1}{2}\mathbb{P}(D|\theta_{SH}) + \frac{1}{2}\mathbb{P}(D|\theta_{SV})} \tag{3}$$

$$\mathbb{P}(c = \mathrm{S}|D) = 1 - \mathbb{P}(c = \mathrm{P}|D) \tag{4}$$

$$\mathbb{P}(D|\theta) = \mathcal{N}\left(\mathbf{z}; 0, \mathbf{K}_{\theta}(\mathbf{x}, \mathbf{x}) + \sigma_p^2 \mathbf{I}\right) \tag{5}$$

where $\sigma_{\mathrm{p}}^2$ is the variance of the participant's (Gaussian) perceptual noise on the pixel value, and $\mathbf{K}_{\theta}(\mathbf{x}, \ \mathbf{x}^{'})$ is a matrix with element $(i, j)$ being $K_{\theta}(x_i, x_j^{'})$ (*Equation 2*). Note that the length scales of the three pattern types as assumed by the observer, $\theta$, need not necessarily be the same as those actually used to generate the images, and indeed we explore below variants of the model that differ

in their assumptions about these length scales. For simplicity, the ideal observer model assumes a fixed extraction of information from each revealing and therefore does not include temporal factors such as fixation durations (i.e. perception noise in *Equation 5* is constant).

## The Bayesian active sensor model of eye movements

The Bayesian active sensor (BAS) algorithm computes a score, the expected reduction in entropy of the distribution over categories as a function of a possible next revealing location, $x^*$, and chooses the next fixation to be at the location with the highest score. The score is defined as

$$\text{Score}(x^*|D) = \text{H}[c|D] - \langle \text{H}[c|z^*, x^*, D] \rangle_{\mathbb{P}(z^*|x^*, D)} \tag{6}$$

where $z^*$ is a possible (and as yet, unobserved) pixel value at $x^*$, and $\text{H}[\cdot]$ denotes entropy in bits. Using the insight that the BAS score formally expresses the mutual information between $c$ and $z^*$ (for the given $x^*$), *Equation 6* can be rewritten in a different form that is computationally far more convenient, as it does not require expensive posterior updates for a continuum of imaginary data, $z^*$, to compute $\mathbb{P}(c|z^*, x^*, D)$ for the second term (*Houlsby et al., 2011*):

$$\text{Score}(x^*|D) = \text{H}[z^*|x^*, D] - \langle \text{H}[z^*|x^*, c, D] \rangle_{\mathbb{P}(c|D)} \tag{7}$$

This form (equivalent to *Equation 1*) is also more plausible psychologically as it is easily approximated by simple mental simulation for a few hypotheses sampled from the current posterior. Note that maximizing just the first term would be equivalent to "maximum entropy sampling" (*Sebastiani and Wynn, 2000*) which is suboptimal in general, and in the context of our task would be similar to simple "inhibition of return" which does not account well for participants' fixations (*Figure 4B*). The two distributions needed for evaluating *Equation 7* are the current posterior over categories, $\mathbb{P}(c|D)$, and the predictive distribution of the pixel value at a location for a category, $\mathbb{P}(z^*|x^*, c, D)$. (Note that the predictive distribution in the first term can also be computed using these two distributions: $\mathbb{P}(z^*|x^*, D) = \langle \mathbb{P}(z^*|x^*, c, D) \rangle_{\mathbb{P}(c|D)}$.) The category posterior is given by *Equation 3-5* and the category-specific predictive distribution is (*Rasmussen and Williams, 2006*):

$$\mathbb{P}(z^*|x^*, c = \text{P}, D) = \mathbb{P}(z^*|x^*, \theta_{\text{PA}}, D) \tag{8}$$

$$\mathbb{P}(z^*|x^*, c = \text{S}, D) = \frac{\mathbb{P}(D|\theta_{\text{SV}})}{\mathbb{P}(D|\theta_{\text{SV}}) + \mathbb{P}(D|\theta_{\text{SH}})} \mathbb{P}(z^*|x^*, \theta_{\text{SV}}, D) + \frac{\mathbb{P}(D|\theta_{\text{SH}})}{\mathbb{P}(D|\theta_{\text{SV}}) + \mathbb{P}(D|\theta_{\text{SH}})} \mathbb{P}(z^*|x^*, \theta_{\text{SH}}, D) \tag{9}$$

with

$$\mathbb{P}(z^*|x^*, \theta, D) = \mathcal{N}\Big(z^*; \mathbf{K}_\theta(x^*, \mathbf{x}) \ [\mathbf{K}_\theta(\mathbf{x}, \mathbf{x}) + \sigma_{\text{p}}^2\mathbf{I}]^{-1}\mathbf{z}, $$
$$K_\theta(x^*, x^*) - \mathbf{K}_\theta(x^*, \mathbf{x})[\mathbf{K}_\theta(\mathbf{x}, \ \mathbf{x}) + \sigma_{\text{p}}^2\mathbf{I}]^{-1}\mathbf{K}_\theta(\mathbf{x}, \ x^*) + \sigma_{\text{p}}^2\Big) \tag{10}$$

For all simulations, we computed the BAS score on a $110 \times 110$ grid of $x^*$ that covered the image. The entropies of the predictive distributions (which are mixtures of Gaussians) in *Equation 7* were approximated by their Jensen lower bound for efficiency (*Huber et al., 2008*).

Importantly, at least in principle, the ideal observer and BAS algorithms can be generalised to more complex stimuli as long as their statistics are known and can be expressed as $\mathbb{P}(z_1, \cdots, z_n|x_1, \cdots, x_n, c)$.

## Saccadic variability and bias

Due to the gaze-contingent design of our experiment, most saccades were made towards locations without any visible target. Saccades are known to be variable and biased, and to incorporate this variability and bias into our model we measured saccadic variability and bias in an independent experiment with 6 participants (including two from the main experiment) under similar conditions to those in the free-scan session of the main experiment. Although in the main experiment participants chose the location they wished to saccade to, to reliably measure saccadic variability and bias we needed to know the target of the saccade but not display it. Therefore we first trained participants on the locations of two saccadic targets and that on each trial the color of a stimulus shown at the fixation point indicated to which of these they needed to saccade. The two targets were at equal eccentricity from the fixation point, one to the right and one above it. On each training trial,

participants first fixated a central fixation cross. Then two isotropic Gaussian patches (SD 0.18° as in the main experiment) appeared, one at the fixation location and one at one of the two target locations. The color of the fixation patch determined where the eccentric target was, either to the right (red) or above (blue), and participants were informed that the relation between the fixation color and the target direction was fixed throughout the experiment. To ensure that participants paid attention to the target, they were asked to report the color of the eccentric target which could either be red or blue. Note that this task by itself did not necessarily require them to saccade to the target, to avoid overtraining on particular saccades, but ensured that they developed a strong association between the color of the central fixation stimulus and the location of the target on that trial, such that in the next phase we could instruct them to saccade to a particular target without showing it. A training session of 40 trials was performed for each target eccentricity tested (1.39, 2.78, 5.56 and 8.34° in this order) with each session followed by a test session of 100 trials with the same eccentricity.

In the test sessions, only the fixation patch appeared and its color determined which target the participant should make an eye movement to. As in the free-scan sessions, a patch was revealed wherever the participant fixated, and their task was to report the target's color. However, as this first eye movement was often inaccurate (see below) it did not necessarily reveal the target which needed to be reported. Thus, to motivate participants by making the task feasible, they were allowed to make four additional eye movements, leading to more revealings, before reporting the color of the target. (Although they were allowed 5 revealings, they were instructed to be as accurate as possible with their first eye movement.) The entire image was then shown as feedback.

To estimate saccadic variability and bias we used only the first saccade after fixation. We removed trials in which participants did not accurately maintain fixation (drift of >0.56°) or where they clearly mis-directed their saccade (saccade was closer to the non-cued target). We calculated the bias and standard deviation (robust estimate from the median absolute deviation) both along (tangential) and orthogonal (SD only) to the direction of the target. We fit the bias and SD as a linear function of target distance for both the tangential and orthogonal components for each participant (*Figure 3—figure supplement 2*). We averaged the model parameters across the participants and used these values to corrupt the desired saccade locations in the BAS simulations (unless otherwise noted).

## Model parameters and data fitting

To match empirical data collected from our participants, we included perception noise, decision noise, and potential prior biases in the ideal observer model.

- To model perception noise, we added Gaussian noise (SD $\sigma_{\mathrm{p}}$) to the displayed pixel values to obtain z, which was either fit to individual participants' categorization data (see below), or set at $\sigma_{\mathrm{p}} = 0.17$ for computing the BAS score (in *Equations 5 and 10*) when determining ideal BAS revealings in passive revealing sessions. The ideal observer model then received these noisy pixel values as input instead of the pixel values actually shown to the participants.

- To incorporate decision noise, the category posterior of *Equation 3-4* was transformed by a softmax function to obtain the probability of choosing the patchy category:

$$\hat{\mathbb{P}}(c = \mathrm{P}|D) = (1 - \kappa)\,\frac{1}{1 + e^{-\beta\,\mathrm{LPR}(D)}} + \frac{\kappa}{2} \tag{11}$$

  where $\kappa$ describes the stimulus-independent decision noise and can be interpreted as the lapse rate, $\beta$ is the stimulus-dependent decision noise (larger values of $\beta$ result in more deterministic behavior and lower values in more random behavior), and $\mathrm{LPR}(D) = \log \frac{\mathbb{P}(c=\mathrm{P}|D)}{\mathbb{P}(c=\mathrm{S}|D)}$ is the log posterior ratio under the ideal observer model (*Equation 3–4*).

- To model prior biases, i.e. imperfect knowledge of the stimulus statistics, we considered three qualitatively different ways in which participants could misrepresent the length scales used to generate the stimuli (6 length scales for three types of stimulus and 2 directions):
  1. no prior biases, i.e. the length scales used by the model were identical to those actually used to generate the stimuli;
  2. a uniform scaling of all length scales relative to their true values, $\alpha$;

3. a fixed offset of all length scales from their true values, $\Delta$;
   Thus, the last two models has a single parameter controlling the relation between the extent of misrepresentation.

For a systematic model comparison, we constructed a set of models which all included perception noise ($\sigma_p$) and stimulus-dependent decision noise ($\beta$) and differed in whether they also included stimulus-independent decision noise ($\kappa$) and which of the three prior biases they had (none, $\alpha$, or $\Delta$). This gave six models and for each we fit all the free parameters on the combined category choice data (from both the free-scan and passive revealing conditions) of each participant using maximum likelihood.

In order to fit the models to empirical data, we needed to take into account that the ideal observer was conditioned on the *perceived* pixel values, which were noisy versions of the pixel values actually displayed (corrupted by perception noise) and were thus unknown to us. Thus, we integrated out the unknown perceived pixel values in order to compute the actual choice probabilities predicted by the model (*Houlsby et al., 2013*). This was approximated by a Monte Carlo integral, by simulating each trial 500 times, drawing random samples of the perceived pixel values given the displayed pixel values from the perceptual noise distribution for each revealing. In each simulated trial, we then computed the probability of a participant's choice according to *Equation 11*, and finally averaged over simulations to obtain the expected probability of each response category in that trial. For optimizing the values of $\sigma_p$ and the parameters for prior bias, $\alpha$ and $\Delta$, we conducted a grid search with a resolution $0.1$, $0.1$, and $0.01°$, respectively. For each setting of these parameters, we optimized $\beta$ and $\kappa$ (when used) using gradient-based search so that all the parameters were jointly optimized. *Table 1* shows the best fit parameters. We used the Bayesian information criterion (computed across all participants) to compare the models by controlling for their differing number of free parameters and chose the best one (*Table 2*).

We used the BAS algorithm to predict the pattern of eye movements for each participant individually given the values of $\sigma_p$ and $\Delta$ fitted to their categorization choices and the measured saccadic inaccuracies. This meant that once we fitted categorization choices, eye movements were predicted without any further tuning of parameters. Note that $\beta$ and $\kappa$ affected only choice probabilities in the categorization decision process, not the eye-movement selection process described by *Equation 1*.

## Revealing densities

As the statistics of stimuli were translationally invariant, the critical features of the optimal solution was the relative locations of the revealing to each other. Therefore, for each trial we first shifted all the revealing locations in that trial equally so that their mean (centre of mass) was located at the centre of the image, thereby removing the variability caused by which part of the image the participants explored in that trial, which was irrelevant to optimality. We then computed the mean density maps by assigning the revealing locations to centers of a $770 \times 770$ grid of bins (the resolution of the images), normalizing the counts, and smoothing the distribution with an isotropic Gaussian filter (std. of 20 bins, ie. $\sim 0.73°$). For a balanced comparison, we computed densities as if each trial had 25 revealings and the three underlying patterns were chosen with equal probability. To achieve this we multiplied the count for the $n^{th}$ revealings by both the relative frequency of first revealings to $n^{th}$ revealings and by the inverse of the frequency of the image type. For each participant, we computed the mean density map across all trials (*Figure 3A* first column) and the mean-corrected density map for each underlying image type (*Figure 3A* last three columns) by removing the mean from those averaged for each image type. The density maps for BAS were generated with simulations that used the same trials (image and number of revealings) that the participants performed. We repeated the simulation of each trial 10 times to obtain a reliable Monte Carlo estimate of the BAS revealing density maps marginalized over the unknown perceived pixel values.

## Correlation analysis

To examine whether the participants' pattern of eye movements depended on the underlying image, and hence what they saw at the revealing locations, we compared correlations between mean-corrected density maps within and across image types for each participant. In computing the correlation as a function of revealing number, we kept the number of samples used to construct the maps constant. This number of samples was chosen to be the maximum number that still allowed the image

type and revealing number to be sampled with equal probability without replacement. To increase the sample size for all revealings we only examined revealings from 5 onwards. For the bar plots, we used about 3 times the number of samples, but weighted each revealing so that the maps were still effectively constructed with an equal number of samples from each revealing number.

To obtain statistics of the correlations, we split the revealing locations for an image type randomly into two data sets of equal size. This produced 6 different data sets (two for each image type). For within-type correlations, we computed the three correlations (using a Gaussian smoothing kernel with $1.5°$ SD) between the mean-corrected maps of the same image types, and averaged these three correlations across the three image types to obtain a single within-type correlation value. Similarly, for across-type correlations, we computed the six correlations between the mean-corrected maps of different image types (across the two different halves of the split), and averaged these six correlations to obtain a single across-type correlation value. We repeated this procedure 1000 times, using a different random split of the data each time, to obtain the means, SD, and 95% confidence intervals (*Figure 3B*).

We also performed the same analysis on the three participants' pooled data, treating the ensemble as data from an 'average' participant. Here the data sets used to calculate the density maps were thus three times larger than those for the individuals. These results together are shown as *participant-self* correlations (*Figure 3B*, left). We applied the same approach to compute *participant-BAS* correlations. For this, participant- as well as BAS-generated revealing location data were each randomly split in half (as above), and correlations were always computed between a participant- and a BAS-generated data set (*Figure 3B*, right).

The correlation between the eye movement patterns derived from correct / incorrect trials and that from BAS was calculated in the same way described above, but only for the 'average' participant as the number of incorrect trials was limited. The p-value reported is the fraction of bootstrapped samples that satisfied the condition $\rho_{\text{correct}} \leq \rho_{\text{incorrect}}$.

## Information gain and efficiency

According to the ideal observer, at any point in a trial, the information gain associated with the set of revealings made thus far is defined as $1 - \mathrm{H}[c|D]$. As described above (Model parameters and data fitting), since we only knew the displayed but not the perceived pixel values, we used Monte Carlo integration to marginalize out the unknown perceived pixel values. To do this, we simulated each experiment 200 times using the parameters of the best fit model, drawing new samples of perceived values for the chosen revealing locations given the displayed pixel values, and then averaged the information gains across runs and trials. *Figure 5A* shows the average information gain as a function of revealing number across the 200 simulations with the average across-trial SEM for each revealing strategy. To characterize the efficiency of each strategy, $s$, we fit the information vs. revealing number curves from each simulation obtained with the above analysis using a cumulative Weibull distribution: $I_s(n) = 1 - \exp\left(-\left(\frac{n}{a_s}\right)^b\right)$, where $n$ is the revealing number, $b$ is a shape parameter shared across strategies (free-scan, passive random, passive ideal BAS, and simulated BAS), and $a_s$ is a strategy-specific scale parameter, which captures the overall efficiency of the strategy. As a relative measure of efficiency for comparing any two strategies, we computed the ratio of their $a_s$, and obtained 95% confidence intervals by using the 200 simulations as bootstrap samples.

## Heuristics

To address whether a heuristic algorithm could account for participants' eye movement patterns, we computed the information gains achieved using several heuristics.

1. **Posterior-independent & order-dependent fixations** (*Figure 5B*, orange). For each participant, the fixation location on the $i^{\text{th}}$ revealing was obtained by sampling (with replacement) from fixation locations pooled across their $i^{\text{th}}$ revealing of all free-scan trials, regardless of the underlying image pattern and the ensuing posterior. This feed-forward strategy respects the participant's average order-dependent fixation map, but is otherwise based on a set of predetermined fixation locations rather than on what was observed about the actual scene.

2. **Posterior-dependent & order-independent fixations** (*Figure 5B*, purple). For each participant, the fixation location on the $i^{\text{th}}$ revealing was obtained by first computing the likelihood $\mathbb{P}(D|\theta)$, where $D$ included the $i$ revealing locations and pixel values observed up to this

revealing, finding the type that had the maximum posterior probability, and then sampling from the fixation locations pooled across trials with that image type as the underlying image pattern. This strategy uses the observations only indirectly, through the posterior, but does not otherwise take into account previous fixation locations and the corresponding pixel values for evaluating the informativeness of potential new revealing locations.

3. **Posterior- & order-dependent fixations** (*Figure 5B*, brown). This combined order-dependence, as in the 1st heuristic, with posterior-dependence, as in the 2nd, but still did not take into account previous fixation locations and the corresponding pixel values. While this strategy would be optimal for simpler stimuli, in which pixels are independent for each image category, or simpler tasks, such as visual search, in which knowledge of the task-relevant posterior (eg. target location) is sufficient for optimal action, it is suboptimal with the kind of naturalistic stimuli and task we used.

## Acknowledgements

We thank C Rothkopf for useful comments on the manuscript, N Houlsby and F Huszár for their contributions to the theory underlying BAS, and J Ingram for technical support. This work was supported by the Wellcome Trust (SC-HY, ML, DMW), the Human Frontier Science Program (DMW), and the Royal Society Noreen Murray Professorship in Neurobiology (to DMW).

## Additional information

### Funding

| Funder | Author |
| --- | --- |
| Wellcome Trust | Scott Cheng-Hsin Yang<br>Máté Lengyel<br>Daniel M Wolpert |
| Human Frontier Science Program | Daniel M Wolpert |
| Royal Society | Daniel M Wolpert |

The funders had no role in study design, data collection and interpretation, or the decision to submit the work for publication.

### Author contributions

SC-HY, Conception and design, Acquisition of data, Analysis and interpretation of data, Drafting or revising the article; ML, DMW, Conception and design, Analysis and interpretation of data, Drafting or revising the article

### Author ORCIDs

Máté Lengyel, http://orcid.org/0000-0001-7266-0049
Daniel M Wolpert, http://orcid.org/0000-0003-2011-2790

### Ethics

Human subjects: The study was approved by the Cambridge Psychology Research Ethics Committee. All participants gave written informed consent prior to the experiment.

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
