## [Decision Letter]

Thank you for submitting your work entitled "Active sensing in the categorization of visual patterns" for consideration by *eLife*. Your article has been favourably reviewed by three peer reviewers, one of whom is a member of our Board of Reviewing Editors. The evaluation has been overseen by this Reviewing Editor and Eve Marder as the Senior Editor.

The reviewers have discussed the reviews with one another and the Reviewing Editor has drafted this decision to help you prepare a revised submission.

Having conferred about this paper, we agree that it is acceptable for publication pending revision. However, there was an extended discussion among the reviewers about the extent to which the current results apply to natural vision. The reviewers agree that the authors need to qualify their conclusions to clearly state that while they were trying to emulate natural vision, their task was unnatural in several important ways (e.g. very small exposures, uniform statistics, small saccades not allowed, and reviewing for up to 60s after presentation in the passive conditions), all of which may have changed the participants' strategy from that used in natural vision. Natural vision works under entirely different constrains – larger exposures, natural scene statistics, all saccades allowed, and limited time. Under these constrains the selection of targets may (or may not) be completely different. These differences between the paradigm used in the study and natural vision should be carefully discussed, and relevant parts of the Abstract and Discussion should be modified accordingly.

[Editors’ note: a previous version of this study was rejected after peer review, but the authors submitted for reconsideration. The previous decision letter after peer review is shown below.]

Thank you for choosing to send your work entitled "Active sensing in the categorization of visual patterns" for consideration at *eLife*. Your full submission has been evaluated by Eve Marder (Senior editor) and three peer reviewers, one of whom is a member of our Board of Reviewing Editors, and the decision was reached after discussions between the reviewers. Based on our discussions and the individual reviews below, we regret to inform you that your work will not be considered further for publication in *eLife*.

While all of the reviewers appreciated the significance of the work, they were very concerned about potential confounds due to letting subjects continue viewing the display for 60s after the last revealing. Given this confound, the reviewers were not certain that a revision would adequately deal with this problem, which led to the decision to reject the paper at this time. If you feel you can adequately answer the concerns of this review, the reviewers were sufficiently potentially positive about your manuscript that *eLife* would allow a submission of a new manuscript that addresses these concerns.

*Reviewer #1:*

Summary

The present work utilizes an innovative gaze-contingent paradigm where subjects have to identify the category of a texture and can actively reveal locations based on where they saccade to. The authors compare the fixation density maps for the three different stimulus categories and find that they are consistent and category-specific across participants, i.e. each stimulus category evokes a specific pattern of fixations. When revealing locations randomly controlled by the computer the categorization performance decreases indicating that active sensing does increase the information yield for the participants. Next, they construct an ideal observer model that computes the posterior probability of each category given the revealed pixel values, and fit the parameters prior bias, perception noise and decision noise to predict the subject's choice of category. Based on these parameters fitted to the subject's category choice the authors try to predict saccade choices using a Bayesian active sensor algorithm which chooses the saccade location which maximizes the expected information gain, taking into account saccade inaccuracies. They claim that the subject's saccades are predicted well by their Bayesian active sensor algorithm, as the predicted category-specific fixation density maps are significantly correlated with the actual density maps. As a better measure of the goodness of their model they compare the information (in the ideal observer sense) gained by the subject's saccades to the information gained by their simulated saccades, random saccades and saccades based on other strategies. They find that the information gained by the subject's saccades is higher than random saccades and saccades based on three heuristic strategies but lower than the information of the saccades simulated by the Bayesian active sensor, even if prior bias, perception noise and category decision noise are taken into account. They call the latter discrepancy a 30% inefficiency of saccade choice. To show that the saccade choices made by the subject are indeed not optimal in efficiency of making the category choice, they reveal the fixation locations computer-controlled based on the optimal locations predicted by the active Bayesian sensor model without biases and low noise. Thereby, the subject's categorization performance is increased. The efficiency of saccade choice does not vary over time, so there does not appear to be any learning.

Remarks

1) In Figure 1, it is not clear to me, why the patch surrounded by the red circle should have more information about zebra vs. cheetah than other blue patches, so maybe use a better example.

2) In Figure 2, the blue condition titled Random should be called passive or computer-controlled random locations to avoid confusion. Also, please clarify the term "gaze-contingent".

3) In Formula 1, D is not defined.

4) Regarding Figure 4, I don't understand why the first term of the formula isn't maximal at locations as far as possible from previous locations (i.e. on the borders of the image). Given that spatial correlations of the stimulus decrease with distance, the uncertainty about pixel values at a far distance from known pixel values should be highest.

5) The correlation analysis done in Figure 3 should be repeated with simulated entropy-maximizing saccades, to show that the correlation analysis is meaningful. Just correlating the fixation density maps might not be the best measure to evaluate fit of the simulation, since it does not consider each fixation on a case-by-case basis given the prior information from previous saccades.

6) In addition to the BIC it would be nice to know the cross-validated prediction accuracy of category choice on a testing set.

7) Also, when showing the performance decrease when revealing at random locations, I would have matched the inter-saccadic distribution (including the variance) and the saccadic distance distribution, to ensure the performance decline is not due to the variance in fixation duration or the subject needing to recover from longer saccades.

*Reviewer #2 (General assessment and major comments (Required)):*

The paper addresses the strategy underlying the selection of gaze targets along the scanpath of fixations. Specifically, the paper compares human selections and performances with those of an optimal Bayesian algorithm. The authors conclude that while humans employ an information-based strategy, this strategy is sub-optimal. They also identify the sources of this sub-optimality and suggest that "participants select eye movements with the goal of maximizing information about abstract categories that require the integration of information from multiple locations".

This is potentially an important paper, which can contribute significantly to the understanding of perceptual processes. The topic is important, the work is, for the most part, elegant and the paper is in general well written. However, there are several crucial points that question the conclusions reached by the authors – these and other comments are listed below.

1) The authors employ a reductionist method, which is indeed necessary if one wants to reveal underlying mechanisms. However, some of the reductionist choices made in this work fuel the questioning of several of its conclusions. The most concerning reductionist steps are:

A) Small saccades (< 1 deg) were not allowed. The use of small ("micro") saccades is task dependent and it may be the case, for example, that with images like those used here a possible strategy is scanning a continuous region. This possibility was "reduced-out" here.

B) The exposed area was effectively < 0.5 deg and was scaled down in transparency from the center out – this reduced-out a meaningful drift-based scanning of the fixational region.

C) The images had uniform statistics, which preclude generalization to natural images.

The authors mention that "the task still allowed them to employ natural every-day eye movement strategies as evidenced by the inter-saccadic intervals (mean 408 ms) that were similar to those recorded for everyday activities […]" – evidently, normal ISIs do not indicate a natural strategy – a) there is much more into an active strategy and b) ISIs seems to be determined by hard-wired circuits that are only slightly modulated in different contexts – see a variety of tests of ISIs in the last decade.

2) The "last 60s" problem. Natural viewing involves a scanpath of fixations. However, natural viewing does not allow a re-scanning of the collection of previous fixation locations. The design of the trial is thus odd– why were the subjects given those extra 60s at the end of the trial to rescan their revealings? This is not justified in the paper and it is actually hidden, in a way, in the Materials and methods section – it took me some time to understand that this was the case. The implications of this design are significant:

A) How can the authors relate performance to either of the two different phases of the trial (the "acquiring" and the "rescanning")?

B) Even in the "acquiring" phase, how can the authors rule out brief rescanning of exposed revealings for periods shorter than the threshold period?

C) If performance depended on the perception during these 60s, then what was the underlying strategy while rescanning?

The authors must describe the procedure clearly at the outset, relate to all these issues explicitly, and explain how they affect (or not) their conclusions.

3) As indicated by the above, the primary characterization of the active strategy here is about *which* revealing were or should be selected, and less about when or in what order. True, order was a factor during the acquiring phase, but not during the rescanning phase. Order seems to play a role in the correlations in Figure 3 but it is not clear by how much. The collection of revealing locations may indeed be the most relevant factor when trying to categorize an image out of several known images with known structures. This is, however, not the case in most natural cases, in which real-time information is essential.

4) From the above and from further analysis it appears that the strong statements of the Abstract are not supported by the data in the paper.

A) "Categorization performance was markedly improved when participants were guided to fixate locations selected by an optimal Bayesian active sensor algorithm […]" – this statement sounds as if guiding the next saccade improves performance. This cannot be concluded here because of the "60 s problem" – during rescanning subjects could select any pattern they wanted. Moreover, performance may even depend on the rescanning period *only*.

B) "By using […] we show that a major portion of this apparent suboptimality of fixation locations arises from prior biases, perceptual noise and inaccuracies in eye movements[…]" – you do not really *show* this. You show that if you add these imperfections to your model you get performance level that resembles that of the subjects. However, there are so many ways to impair the performance of your model – where do you show that these are the crucial factors?

C) "The central process of selecting fixation locations is around 70% efficient" – this is correct *only* for the specific task and context of your experiment. The statement sounds much more general than that.

D) "Participants select eye movements with the goal of maximizing information about abstract categories that require the integration of information from multiple locations" – again, a statement with a general flavor without such justification – the statement should be toned and tuned down to reflect the narrow context of the findings.

5) Audience and style. As written now the paper seems to address experts in Bayesian models of perception. Given that it is submitted to *eLife* I assume that the paper should address primarily, or at least to the same degree, biologists who are interested in understanding perception. And indeed, papers like this form a wonderful opportunity to create a productive dialogue between biologists and theoreticians. For this end, the style of writing must change. The amount of statistical details should be reduced, the biological meaning of the various strategies discussed should be provided, the rational for selecting the BAS algorithm as the optimal strategy should be explained, the meaning of each strategy in terms of actual eye movements in natural scenes should be explained. At the end, the reader should understand the biological meaning of each strategy (that is, a biologically-plausible description of the strategy employed by their subjects in this task, and a strategy they would employ in a natural task), the rational of why one strategy is better than the other, and in what sense humans are sub-optimal.

6) The discussions about active-sensing in the paper are written as if they come from a sort of a theoretical vacuum. Active sensing has been introduced, studied and discussed at various levels for various species and modalities for years, and intensively so for the last decade or two. Nevertheless, the paper sounds as if these concepts are only beginning to be addressed, and as if active vision is mostly covered by addressing saccades scanpath selections. This introduces a huge distortion of the concepts of active sensing and active vision. Active vision is much more than scanpath selections. Eye movements include saccades and drifts, and as far as we know today vision does not occur without the drift movements. Saccades include large and small ("micro") saccades, making a continuous spectrum of saccade amplitudes. In this study only large (> 1 deg) saccades were allowed. Thus, this study ignores a substantial portion of visually-crucial eye movements. While this is ok as a reductionist method, it is not ok to ignore the reduced-away components in the discussion and interpretation of the results. Thus:

A) There is no justification to use the term "eye movements" in this paper – the components of eye movements studied here should be termed properly (perhaps use terms like "fixation scanpath" or "saccadic order" or anything else that is appropriate).

B) Previous results and hypotheses about active sensing/active vision that refer to all kinds of movements must be discussed and referred to when interpreting the current results.

C) The results of the current study should be put in the context of this general active sensing context, and the generality of the conclusions should be phrased accordingly – mostly, they should be toned down and related to the specific reduced context of this study.

Minor comments:

1) Arbitrary selection of parameters – please explain the choice of every parameter you use (e.g., the criteria for saccades and fixations).

2) Figure 3 – please run a shuffling control analysis to show how much of the correlation is order-dependent.

*Reviewer #3:*

The authors investigated the control of eye movements in a visual categorization task. By using a gaze-contingent paradigm, they could show that eye movement patterns became more specific to the underlying image structure with increasing fixation number. A comparison with a Bayesian active sensor (BAS) model showed that humans were quite efficient, given certain biases and inaccuracies of eye movements.

The study should be interesting to a broad audience and is well conducted. A few further analyses could strengthen the message.

1) The authors corrected for errors in saccade targeting tangential and orthogonal to saccade direction. However, as they correctly stated in the Discussion, there are also other potential sources of errors due to biases along cardinal directions or the bias to fixate the center of an image. It would be very interesting to see how much of the performance reduction is caused by these biases. It should be possible to assess these biases directly from the existing data.

2) Since there were a lot of incorrect judgments overall, it would be interesting to compare the density maps for correct and incorrect judgments. If eye movements are indeed controlled by an active strategy, the differences and correlations should be more pronounced for correct judgments.

3) The example trial in Figure 4 provides a good illustration of BAS and the maximum entropy variant but it does not allow a quantitative comparison. It would be helpful to show the full distributions of percentile values with respect to BAS and entropy scores.

4) As I understand the task, the revealings stayed on screen after the last revealing for 60 s or an unlimited duration, depending on the condition. The behavior of the subjects after the last revealing should be reported because it could have influenced their perceptual judgment.

---

## [Author Response]

*Having conferred about this paper, we agree that it is acceptable for publication pending revision. However, there was an extended discussion among the reviewers about the extent to which the current results apply to natural vision. The reviewers agree that the authors need to qualify their conclusions to clearly state that while they were trying to emulate natural vision, their task was unnatural in several important ways (e.g. very small exposures, uniform statistics, small saccades not allowed, and reviewing for up to 60s after presentation in the passive conditions), all of which may have changed the participants' strategy from that used in natural vision. Natural vision works under entirely different constrains – larger exposures, natural scene statistics, all saccades allowed, and limited time. Under these constrains the selection of targets may (or may not) be completely different. These differences between the paradigm used in the study and natural vision should be carefully discussed, and relevant parts of the Abstract and Discussion should be modified accordingly.*

Thank you for the reviewers’ comments on our revised manuscript. We have now added in a whole section in the Discussion on the relation of our task and results to natural vision and discuss in detail the four issues raised by the reviewers. Some of these issues were discussed already in other parts of the paper such as the Methods but have now been put together in this section headed “Relevance for natural vision”.

[Editors’ note: the author responses to the previous round of peer review follow.]

*While all of the reviewers appreciated the significance of the work, they were very concerned about potential confounds due to letting subjects continue viewing the display for 60s after the last revealing. Given this confound, the reviewers were not certain that a revision would adequately deal with this problem, which led to the decision to reject the paper at this time.*

All three reviewers were concerned that allowing additional time after the last revealing (rescanning period) was a confound in the experiment and undermined our conclusions. In fact the additional time is essential for the interpretation of the experimental results and does not affect our conclusions. We realise we did not highlight this sufficiently and have now done so in the revised manuscript.

1) As we wished to compare the active strategy with passive revealing we allowed participants to rescan the revealed locations after the final revealing but before making a decision. This was critical as in the passive revealing conditions, although participants may detect and follow the revealings, because they are small and dispersed, the participants need to scan the scene to find and view them all. Therefore, the reason for allowing additional time after the final revealing was so as to make sure they had a chance to extract as much information as they like from the revealed locations (locations and pixel values). In order to make the conditions directly comparable, we followed the same procedure in the active condition. Thus, allowing a rescanning period was the only way we could make a fair comparison across conditions using our gaze contingent design, which in turn allowed us fine control over the information participants could obtain by eye movements.

2) Critically, our key interest is in where participants choose to fixate during the initial revealing period and not in the perception model. During the active condition all the selection happens prior to the rescanning phase and participants do not know how many saccades they will be allowed so they still need to be efficient in revealing locations even if they can rescan them later. Therefore, as we are analyzing the initial selection, the final rescanning simply equalizes the information that can be extracted from each revealing for all conditions but it is unlikely to influence the initial selection. Although, it is theoretically possible that participants adopted a different eye movement strategy knowing they could freely re-visit already revealed locations, given the little improvement they showed across sessions in our task (Figure 7), this remains a highly unlikely possibility.

3) Although we allowed up to 60 s after the final revealing (we wished participants to feel that they had as long as they wanted), participants took on average only 5.0 s to make a decision in the active condition, 6.4 s in the passive random and 4.4 s in the passive BAS. We expected that participants would use rescanning time so that information extracted from the revealings would have saturated by the time of choice. To examine whether rescanning time had an effect on performance, we fit each participant’s choice accuracy as a logistic function (bounded between 0.5 and 1) of rescanning time. We allowed different shifts per condition (active, passive random, passive BAS) but the same slope parameter across conditions. This showed that for two participants there was a significant effect of rescanning time on performance (i.e. slope significantly different from zero) but with a decrement in performance for longer rescanning times. The probability of a correct decision for rescanning times at the 25th and 75th percentiles of the rescanning time distribution falls from 0.77 to 0.54 (p<0.001) and from 0.77 to 0.66 (p <0.03) for these two participants. We have included this information in the revision. Therefore, if anything, participants did not use longer rescanning times to improve performance but may have taken a little extra time when they were uncertain. In contrast, their performance correlated quite strongly with the number of revealings (Figure 2). This indicates that the main determinant of their performance was how well they chose the revealing locations in the first place, rather than how long they rescanned the location, as our original results already showed.

4. To fully assuage the reviewers’ concerns we have now run a control experiment on 3 new participants in the active revealing condition except that no rescanning after the final revealing was allowed (all revealings returned to a black screen 350 ms after the saccade away from the final revealing). The results from this show that the revealing density maps are very similar to those from the original experiment (average within-type vs. across-type correlation across the two groups of participants: 0.63 vs. 0.30) and performance is also similar (although not surprisingly slightly worse). The proportion correct across all active revealing trials for the original participants was 0.65, 0.66 & 0.69 (average 0.66) and for the new controls 0.64, 0.58 & 0.66 (average 0.63). We include these new results in the paper and in a supplementary figure (Figure 2—figure supplement 1) and take them to indicate that allowing rescanning in our original design did not change participants’ revealing strategy.

In addition, we realized we could further improve our analysis of the revealing density maps. In particular, in our original analysis these maps were constructed using the absolute positions of revealings. However, the statistics of our stimuli are translationally invariant and therefore the critical features of the optimal solution are the *relative* locations of the revealing to each other. Therefore, for each trial we first shifted all the revealing locations in that trial equally so that their mean (centre of mass) was located at the centre of the image, thereby removing the variability caused by which part of the image the participants explored in that trial, which is irrelevant to optimality (due to the translationally invariant nature of the statistics of our images, see above). This analysis leads to much cleaner and more compelling maps and we have redone all analyses (correlations of revealing densities) based on these new densities. We checked that all conclusions (including those reported in the response to reviewers below) remained the same (although quantitatively stronger) compared to the original absolute location method.

Reviewer #1:

*Remarks*

*1) In Figure 1, it is not clear to me, why the patch surrounded by the red circle should have more information about zebra vs. cheetah than other blue patches, so maybe use a better example.*

We have modified this figure and legend to make the example more intuitive.

*2) In Figure 2, the blue condition titled Random should be called passive or computer-controlled random locations to avoid confusion. Also, please clarify the term "gaze-contingent".*

We have relabelled Figure 2 and Figure 5 to explicitly show the passive and active conditions and modified the text to use consistent terminology where necessary. We have also added a phrase to clarify the term “gaze-contingent.”

*3) In Formula 1, D is not defined.*

D was actually defined in the previous section “Bayesian ideal observer” already, but we now define it again right after Eq. 1.

*4) Regarding Figure 4, I don't understand why the first term of the formula isn't maximal at locations as far as possible from previous locations (i.e. on the borders of the image). Given that spatial correlations of the stimulus decrease with distance, the uncertainty about pixel values at a far distance from known pixel values should be highest.*

We understand that this seems counter-intuitive, but it is correct. We try here to give an intuition. However, as the maximum entropy model is not the focus of the paper we would prefer to omit the lengthy description and figure but will be happy to include them in the paper if the Reviewer deems it necessary.

We have added a short note to the caption of Figure 4 to alert the reader to the non-intuitive nature of the MaxEnt model, and also included a new Figure 4—figure supplement 1.

In more detail, the total uncertainty (equivalent to the first term on the right hand side of Eq. 1) about pixel value can be decomposed into two parts (this can be most easily understood formally if entropy is substituted by variance, by the law of total variance, Figure 4—figure supplement 1.). The first part is “unexplained variance” (equivalent to the second term on the right hand side of Eq. 1): even if we knew what the image type was, there is still uncertainty due to the stochastic nature of the stimulus and perception noise. Unexplained variance *increases* with distance from revealings. Because the stimulus is spatially correlated, i.e. it is not white noise, knowing about the pixel value at a particular revealing location provides some information about nearby pixel values but not about distant pixel values. The second part is “explained variance” (equivalent to our BAS score, Eq. 1) and it is due to uncertainty about image type and the fact that each type predicts different pixel values at the characteristic length scale of stimulus spatial correlations. At very short length scales, each image type is bound to predict the same pixel values as that at the revealing location (as they are constrained by this observation), and so the explained variance is diminishingly small, and beyond the stimulus correlation length scale the differences again become small as the average predictions are the same (zero) due to stochasticity. Thus, explained variance peaks around the stimulus correlation length scales (which in our case are small relative to the total image size) and fall off from there, resulting in a contribution that predominantly *decreases* with distance from revealings. Therefore, the total uncertainty determined as a combination of these two sources of uncertainty can peak at the borders or at intermediate distances, close to the autocorrelation length scale of the stimulus, depending on the balance of these two sources, which in turn depends on the revealed pixel values (and locations)

*5) The correlation analysis done in Figure 3 should be repeated with simulated entropy-maximizing saccades, to show that the correlation analysis is meaningful. Just correlating the fixation density maps might not be the best measure to evaluate fit of the simulation, since it does not consider each fixation on a case-by-case basis given the prior information from previous saccades.*

We repeated the correlation analysis simulating the maximum entropy strategy. The results show no significant correlation between the revealing locations chosen by MaxEnt and those chosen by the participants (Author response Table 1). This suggests that the correlation shown in the original Figure 3 is meaningful and that MaxEnt does not describe our participants’ behaviour well. We now present this analysis in the results but would prefer not to overload the paper with this additional Table.

within-type correlation;p-value for correlation (ρ)across-type correlation; p-value for correlation (ρ)participant 1ρ=-0.121; p=0.13ρ=0.060; p=0.13participant 2ρ=-0.117; p=0.12ρ=0.056; p=0.13participant 3ρ=-0.024; p=0.43ρ=0.017; p=0.38

Author response table 1. Correlation between participant’s eye movement and those derived from the MaxEnt algorithm at 25 revealing.

We also agree that the correlation is not the best way to evaluate the model but they are intuitive to understand. Critically, this is why we also analyzed the information curves which are a more principled measure that includes prior information from previous saccades.

*6) In addition to the BIC it would be nice to know the cross-validated prediction accuracy of category choice on a testing set.*

We have now performed this analysis and the results are in Author response table 2. We computed 10-fold cross validated prediction errors by 10 times holding out a different random 10% of the data, fitting parameters to the remaining 90% and measuring prediction performance on the held out 10%. As one can see, cross validation errors show the same trend as BIC values, and thus we decided not to include this in the manuscript as the BIC is more informative for model comparison.

BIC difference from Table 2:Average prediction error per trial (arithmetic mean; geometric mean):1600.4294; 0.42621390.3958; 0.4246580.3671; 0.419400.3658; 0.41791050.3731; 0.42371020.3729; 0.4228

Author response table 2. BIC values and cross-validated prediction errors.

*7) Also, when showing the performance decrease when revealing at random locations, I would have matched the inter-saccadic distribution (including the variance) and the saccadic distance distribution, to ensure the performance decline is not due to the variance in fixation duration or the subject needing to recover from longer saccades.*

As discussed above in the response to the editors, in both the passive and active conditions, participants could rescan the revealed locations after the final revealing before making a decision. Therefore, these concerns are not relevant as the participants can fixate as long as they like each of the revealed locations. Please also see the response to the editors above for the rationale for the time that participants are allowed to rescan the scene after the last revealing.

Reviewer #2:

*1) The authors employ a reductionist method, which is indeed necessary if one wants to reveal underlying mechanisms. However, some the reductionist choices made in this work fuel the questioning of several of its conclusions. The most concerning reductionist steps are:*

*A) Small saccades (< 1 deg) were not allowed. The use of small ("micro") saccades is task dependent and it may be the case, for example, that with images like those used here a possible strategy is scanning a continuous region. This possibility was "reduced-out" here.*

*B) The exposed area was effectively < 0.5 deg and was scaled down in transparency from the center out – this reduced-out a meaningful drift-based scanning of the fixational region.*

The reviewer is concerned that micro-saccades and drift could play an important role in our task but that we do not account for them in our experimental protocol. As we now clarify in the revised manuscript, the key aim of the experiment was to look at a more voluntary component of movement, that is, the scan paths of fixations, rather than the more involuntary process of

micro-saccades and drift (Rolfs, 2009). Over the course of each fixation in our experiment (~350 ms), based on the work of Rolfs (2009) who systematically studied fixation variability (ibid Figure 4), the SD of eye position is on the order of 0.22 deg. As each of our Gaussian revealing aperture had a SD of 0.18 degree, drifts and microsaccades were in fact likely to be used for extracting information within each revealed patch. Critically, if we allowed revealings on this length scale, the revealed locations would be too close together to be informative, as the smallest length scale of the our patterns (stripy) is 0.91 deg and is more than 4 times larger than the typical distance covered by microsaccades and drift (0.22 deg, see above).

Finally, if participants are limited in the number of saccades they are allowed to make (micro or otherwise) then our model makes it very clear that micro- (or small) saccades are highly sub-optimal for the high-level category judgments we study, and in agreement with this our participants’ selection of revealing locations were on average 3.51 deg apart. Therefore, we feel that the existence of drift and micro-saccades does not undermine the main message of our paper and include a discussion of these issues, in particular, that they may play a complementary role in increasing information about low-level visual features.

*C) The images had uniform statistics, which preclude generalization to natural images.*

*The authors mention that "the task still allowed them to employ natural every-day eye movement strategies as evidenced by the inter-saccadic intervals (mean 408 ms) that were similar to those recorded for everyday activities […]" – evidently, normal ISIs do not indicate a natural strategy – a) there is much more into an active strategy and b) ISIs seems to be determined by hard-wired circuits that are only slightly modulated in different contexts – see a variety of tests of ISIs in the last decade.*

Natural image statistics are often assumed to be spatially stationary (e.g. Field, 1987), although this is undoubtedly an approximation. Our stimulus was more naturalistic than many of the stimuli used in active sensing studies of visual search (e.g. 1/f noise) while still allowing a rigorous control and measurement of the amount of high-level category information available at any potential revealing location which would have not been possible with real natural scenes. Thus, we felt our stimuli strike the right balance on the eternally debated natural-vs.-artificial stimulus scale. We agree that ISIs by themselves are only necessary but not sufficient to prove that natural eye movement strategies are at play, but – as we also mention in the manuscript – we also found little to no learning across several sessions which we take as further (albeit still only circumstantial) evidence. Importantly, at least in principle, our mathematical approach generalises to more complex stimuli as long as their statistics are known and can be expressed as P(z_1_ z_n_ x_1_ x_n_, c) (with the notation used in the manuscript). We have included these points in the Methods to more clearly express the strengths and limitations of our stimuli and approach.

*2) The "last 60s" problem. Natural viewing involves a scanpath of fixations. However, natural viewing does not allow a re-scanning of the collection of previous fixation locations. The design of the trial is thus odd– why were the subjects given those extra 60s at the end of the trial to rescan their revealings? This is not justified in the paper and it is actually hidden, in a way, in the Methods section – it took me some time to understand that this was the case. The implications of this design are significant:*

*A) How can the authors relate performance to either of the two different phases of the trial (the "acquiring" and the "rescanning")?*

As this point was raised by all three reviewers and highlighted as the key reason the paper was rejected, we have provided a full response to editors above (that includes an analysis of performance as a function of rescanning time as well as a new control experiment without rescanning).

*B) Even in the "acquiring" phase, how can the authors rule out brief rescanning of exposed revealings for periods shorter than the threshold period?*

The revealed locations are generally far apart so it is not possible to return to a revealing without triggering an additional revealing. Even if participants could rescan during the revealing stage this still does not undermine the analysis of whether they are efficient in the selection of N discrete revealings.

*C) If performance depended on the perception during these 60s, then what was the underlying strategy while rescanning?*

Please see point 3 and 4 in the response to the editors for this point.

*The authors must describe the procedure clearly at the outset, relate to all these issues explicitly, and explain how they affect (or not) their conclusions.*

We have now clarified all the issues relating to rescanning in the revision.

3) As indicated by the above, the primary characterization of the active strategy here is about which revealing were or should be selected, and less about when or in what order. True, order was a factor during the acquiring phase, but not during the rescanning phase. Order seems to play a role in the correlations in Figure 3 but it is not clear by how much (see below). The collection of revealing locations may indeed be the most relevant factor when trying to categorize an image out of several known images with known structures. This is, however, not the case in most natural cases, in which real-time information is essential.

This is really the rescanning issue again. As the participant does not know how many saccades they will be allowed on each trial, they have to select them in an order that will be informative at the time they choose them and several of our analyses examine order effects. Therefore the selection process operates in real time. Please see the full response to the editors. We agree that we are not studying the “when” question (i.e. how long each fixation should be hold for) and we mention this in the Materials and methods.

*4) From the above and from further analysis it appears that the strong statements of the Abstract are not supported by the data in the paper.*

*A) "Categorization performance was markedly improved when participants were guided to fixate locations selected by an optimal Bayesian active sensor algorithm […]" – this statement sounds as if guiding the next saccade improves performance. This cannot be concluded here because of the "60 s problem" – during rescanning subjects could select any pattern they wanted. Moreover, performance may even depend on the rescanning period* only.

Please see the response to the editors. We do not claim the participant has to see the revealings in a specific order in the passive condition and our perceptual model does not take order into account (and the reason for this is that we allow rescanning). To reiterate, in order to equate the information on location and pixel color in the passive and active conditions we allow the rescanning period. The performance may well depend on only the rescanning period in both conditions but critically the selection in the active condition has to be done in real time. We have revised the phrasing of this sentence to: “categorization performance was markedly improved when locations were revealed to participants by an optimal Bayesian active sensor algorithm.”

*B) "By using […] we show that a major portion of this apparent suboptimality of fixation locations arises from prior biases, perceptual noise and inaccuracies in eye movements […]" – you do not really show this. You show that if you add these imperfections to your model you get performance level that resembles that of the subjects. However, there are so many ways to impair the performance of your model – where do you show that these are the crucial factors?*

We chose to start with what we (and we expect most readers) would regard as the most natural sources and forms of sub-optimality – sensory and motor noise and biases. Clearly, the number of potential models is unbounded but in any given scientific study one has to consider a finite set of possible alternative models. We did do formal comparison of a set of models, so we emphasize that it is not as though we hand-picked one. Our aim was to explain the data parsimoniously and we are happy to consider alternative models if the reviewer has something particular in mind but do feel the set of models we examined forms a reasonable set for our study. We have removed the word “show” and now use “estimate” instead.

*C) "The central process of selecting fixation locations is around 70% efficient" – this is correct only for the specific task and context of your experiment. The statement sounds much more general than that.*

We clarify that this is for our task. However, note that no paper can ever make a claim of anything apart from what particular task was studied in the experiment. We feel it reasonable that an Abstract then tries to abstract a message for the reader. To take just one example, Najemnik and Geisler in the Abstract of their 2005 Nature paper say “We find that humans achieve nearly optimal search performance” whereas clearly that can’t be stated unless they have examined every search task and stimuli that humans have ever used. But most readers understand what they and others mean by the statement.

*D) "Participants select eye movements with the goal of maximizing information about abstract categories that require the integration of information from multiple locations" – again, a statement with a general flavor without such justification – the statement should be toned and tuned down to reflect the narrow context of the findings.*

We do use the word “estimate” earlier in the Abstract, so again we find this criticism rather unfair as it would apply to almost all papers on this topic.

*5) Audience and style. As written now the paper seems to address experts in Bayesian models of perception. Given that it is submitted to* eLife *I assume that the paper should address primarily, or at least to the same degree, biologists who are interested in understanding perception. And indeed, papers like this form a wonderful opportunity to create a productive dialogue between biologists and theoreticians. For this end, the style of writing must change. The amount of statistical details should be reduced,*

We feel it is very important to back up our results with statistical rigor so would not be keen to remove statistical tests from the results. Indeed other reviewers have asked for more statistical tests which we have included where necessary. If the reviewer means the equations for the BAS, this is the single equation we have in the main text which is so central to the paper we would be reluctant to remove it.

*the biological meaning of the various strategies discussed should be provided, the rational for selecting the BAS algorithm as the optimal strategy should be explained, the meaning of each strategy in terms of actual eye movements in natural scenes should be explained. At the end, the reader should understand the biological meaning of each strategy (that is, a biologically-plausible description of the strategy employed by their subjects in this task, and a strategy they would employ in a natural task), the rational of why one strategy is better than the other, and in what sense humans are sub-optimal.*

We have explained the rationale for selecting the BAS algorithm in the section “Predicting eye movement patterns by a Bayesian active sensor algorithm”. In terms of giving a “biologically-plausible description of the strategy employed by their subjects in this task,” we are a little unsure what the reviewer wants – the detailed description of BAS is the strategy that we propose participants are using, which is to find the location in the scene that when fixated is most likely to lead to the greatest reduction in the categorization error. We now highlight this in words in the Discussion and hope this addresses the reviewer’s comment.

*6) The discussions about active-sensing in the paper are written as if they come from a sort of a theoretical vacuum. Active sensing has been introduced, studied and discussed at various levels for various species and modalities for years, and intensively so for the last decade or two. Nevertheless, the paper sounds as if these concepts are only beginning to be addressed, and as if active vision is mostly covered by addressing saccades scanpath selections. This introduces a huge distortion of the concepts of active sensing and active vision. Active vision is much more than scanpath selections. Eye movements include saccades and drifts, and as far as we know today vision does not occur without the drift movements. Saccades include large and small ("micro") saccades, making a continuous spectrum of saccade amplitudes. In this study only large (> 1 deg) saccades were allowed. Thus, this study ignores a substantial portion of visually-crucial eye movements. While this is ok as a reductionist method, it is not ok to ignore the reduced-away components in the discussion and interpretation of the results. Thus:*

*A) There is no justification to use the term "eye movements" in this paper – the components of eye movements studied here should be termed properly (perhaps use terms like "fixation scanpath" or "saccadic order" or anything else that is appropriate).*

We felt it would be very awkward to replace all mention of “eye movement” in the paper and as suggested we clarify now that we only studied fixation scanpaths and not microsaccades or drift. Again there are many examples in the literature where it is common practice to refer to the kind of study we did as being about “eye movements”. If the reviewer and editors insist, we can replace all mention of eye movements but feel it would make the paper far less accessible.

*B) Previous results and hypotheses about active sensing/active vision that refer to all kinds of movements must be discussed and referred to when interpreting the current results.*

We now reference several papers on microsaccades and drift.

*C) The results of the current study should be put in the context of this general active sensing context, and the generality of the conclusions should be phrased accordingly – mostly, they should be toned down and related to the specific reduced context of this study.*

We have clarified that we study fixation scanpaths.

Minor comments:

1) Arbitrary selection of parameters – please explain the choice of every parameter you use (e.g., the criteria for saccades and fixations).

The parameters were based on pilot experiments on the authors with the aim of making the revealings in the free-scan session feel natural so that each location viewed was revealed and spurious locations were not revealed. We have now explained this in the Methods.

*2) Figure 3 – please run a shuffling control analysis to show how much of the correlation is order-dependent.*

We have performed the shuffling control and in doing this, we realized that the magnitude of correlation depends on the number of samples used to construct the density maps. When we calculated the correlation in our original analysis, the number of samples increased with the number of revealing on the x-axis. This means that the increasing separation could have arisen from the increasing sample size. We have therefore modified the way we calculate the correlation and ensured that the number of samples used to construct the density maps remains the same as the number of revealing increases. The new results are in the updated Figure 3. As one can see, the trend of the correlation curves is the same: the within- and across- type correlations increasingly separate as a function of revealing, with the within-type correlation increasing and the across-type correlation decreasing. In contrast, the shuffling analysis shows that the correlation curves remain constant as a function of revealing number. This is expected for the shuffling case because the density maps are now constructed with random subsamples of the same pool of data regardless of revealing number. The shuffled results are shown in Figure 6 but we don’t think they need to be included in the paper as they have to be flat by construction now that we have ensured that the sample size is independent of revealing number. We thank the reviewer for suggesting the shuffling control as it led to us improving the analysis.

Author response image 1.Correlation as a function of revealing number with the revealing number shuffled.Orange denotes within-type correlation; purple denotes across-type (cf. Figure 3 in the manuscript). Line and shaded area represent mean and SD, respectively.**DOI:**
http://dx.doi.org/10.7554/eLife.12215.016

Reviewer #3:

*1) The authors corrected for errors in saccade targeting tangential and orthogonal to saccade direction. However, as they correctly stated in the Discussion, there are also other potential sources of errors due to biases along cardinal directions or the bias to fixate the center of an image. It would be very interesting to see how much of the performance reduction is caused by these biases. It should be possible to assess these biases directly from the existing data.* It is important to clarify that the suboptimalities the Reviewer asks us to identify are conceptually different from those that we factored out using the specific biases and noise included in the (non-ideal) BAS model. The ones we considered are suboptimalities that arise in the execution of a planned saccade (as in Figure 3—figure supplement 2), while the ones that the reviewer refers to could be suboptimalities of the planning itself. As we are interested in the degree of (sub)optimality of the planning component, we thought it would be misleading to factor out the biases the Reviewer is considering. We realise that our discussion of this issue was confusing in the original manuscript so we have now rewritten it.

The Reviewer asks us to assess possible fixation bias towards the center of the image and saccade directional bias towards cardinal directions. With the old analysis of the density maps that uses the absolute revealing locations (see point 4 in response to the editor), we see that the participants actually tend to fixate away from the center compared to BAS. Similarly, with the new analysis that uses relative locations, we see that the participants actually tend to have a more diffuse distribution of fixation compared to BAS (see mean revealing density maps in Figure 3). The fact that the revealings chosen by BAS are more efficient and more concentrated at the center of the field of view (both in absolute and shifted position) suggests that this type of fixation bias will actually improve the performance rather than reduce it and therefore would not account for any sub-optimality.

Comparing the saccades made by the participants and the BAS algorithm, we see that our participants tend to make more horizontal saccades and fewer vertical saccades (Figure 7). The deviations from BAS may well reduce performance, but it is not obvious how to formalize or determine if this is a true bias on top of planning or part of planning itself. In particular, it is not clear to us how we can measure bias directly from the existing data (as the reviewer suggests) separate from any active strategy as we do not know where subjects are aiming. We would prefer not to add the plot below to the paper as there is no really strong message we can make about it, but are happy to add it if the reviewer wishes.

Author response image 2.Probability density of saccade angles.**DOI:**
http://dx.doi.org/10.7554/eLife.12215.017

*2) Since there were a lot of incorrect judgments overall, it would be interesting to compare the density maps for correct and incorrect judgments. If eye movements are indeed controlled by an active strategy, the differences and correlations should be more pronounced for correct judgments.*

Thank you for this great suggestion. We have now performed this analysis and below are the density maps for correct and incorrect trials (averaged across participants, new Figure 3—figure supplement 3). We chose to analyze only for the “average” participant as the number of incorrect trials is limited for each participant. The correlation with BAS is higher for correct compared to incorrect trials (average of 0.35 vs 0.01; p<0.05). We have added this into the revised manuscript.

*3) The example trial in Figure 4 provides a good illustration of BAS and the maximum entropy variant but it does not allow a quantitative comparison. It would be helpful to show the full distributions of percentile values with respect to BAS and entropy scores.*

We have computed the distribution of score percentiles across revealings for the two sensing algorithms. The result makes it clear that the BAS algorithm is a much better description for human eye movement than the MaxEnt algorithm. We have added this graph as a subpanel of Figure 4 in the revision and mentioned this in the text.

*4) As I understand the task, the revealings stayed on screen after the last revealing for 60 s or an unlimited duration, depending on the condition. The behavior of the subjects after the last revealing should be reported because it could have influenced their perceptual judgment.*

As this point was raised by all three reviewers and highlighted as the key reason the paper was rejected, we have provided a full response to this issue in the response to the editors above.